# Embedding Self-Correction as an Inherent Ability in Large Language Models for Enhanced Mathematical Reasoning

## Abstract

Accurate mathematical reasoning with Large Language Models (LLMs) is crucial in revolutionizing domains that heavily rely on such reasoning. However, LLMs often encounter difficulties in certain aspects of mathematical reasoning, leading to flawed reasoning and erroneous results. To mitigate these issues, we introduce a novel mechanism, the Chain of Self-Correction (CoSC), specifically designed to embed self-correction as an inherent ability in LLMs, enabling them to validate and rectify their own results. The CoSC mechanism operates through a sequence of self-correction stages. In each stage, the LLMs generate a program to address a given problem, execute this program using program-based tools to obtain an output, subsequently verify this output. Based on the verification, the LLMs either proceed to the next correction stage or finalize the answer. This iterative self-correction process allows the LLMs to refine its reasoning steps and improve the accuracy of its mathematical reasoning. To enable the CoSC mechanism at a low cost, we employ a two-phase finetuning approach. In the first phase, the LLMs are trained with a relatively small volume of seeding data generated from GPT-4, establishing an initial CoSC capability. In the second phase, the CoSC capability is further enhanced by training with a larger volume of self-generated data using the trained model in the first phase, without relying on the paid GPT-4. Our comprehensive experiments demonstrate that CoSC significantly improves performance on traditional mathematical datasets among existing open-source LLMs. Notably, our CoSC-Code-34B model achieved a 53.5% score on MATH, the most challenging mathematical reasoning dataset in the public domain, surpassing the performance of well-established models such as ChatGPT, GPT-4, and even multi-modal LLMs like GPT-4V, Gemini-1.0 Pro, and Gemini-1.0 Ultra. It's important to note that, unlike these proprietary models, our CoSC performs inference in a zero-shot manner, without the need for demonstrations. The code and data for this work will be released once this paper is accepted.

## 1 Introduction

Large Language Models (LLMs), such as GPT-4 (OpenAI, 2023a), have recently demonstrated state-of-the-art performance across a variety of natural language processing (NLP) tasks, including natural language generation and understanding (Chowdhery et al., 2023; Team et al., 2023; Anil et al., 2023; Penedo et al., 2023). Despite their success, LLMs often struggle with mathematical reasoning tasks due to their lack of explicit logical reasoning and judgment, which are crucial for solving such problems. Moreover, there is a fundamental gap between natural language and the language of mathematical formulas, which further complicates these tasks. As a result, accurate mathematical reasoning remains an essential yet challenging capability for LLMs to develop, in order to further advance various domains. Consequently, it is still an open challenge to tackle mathematical problems for existing open-source LLMs, such as LLaMA-2 (Touvron et al., 2023).

To improve the mathematical reasoning abilities, numerous approaches have been investigated in previous research, including prompting (Wei et al., 2022; Chen et al., 2023a; Wang et al., 2023b), pretraining (Azerbayev et al., 2023; Fu et al., 2023; Shao et al., 2024), and finetuning (Luo et al., 2023; Yu et al., 2023; Yue et al., 2023; Gou et al., 2023b; Liu et al., 2023b). In particular, finetuning

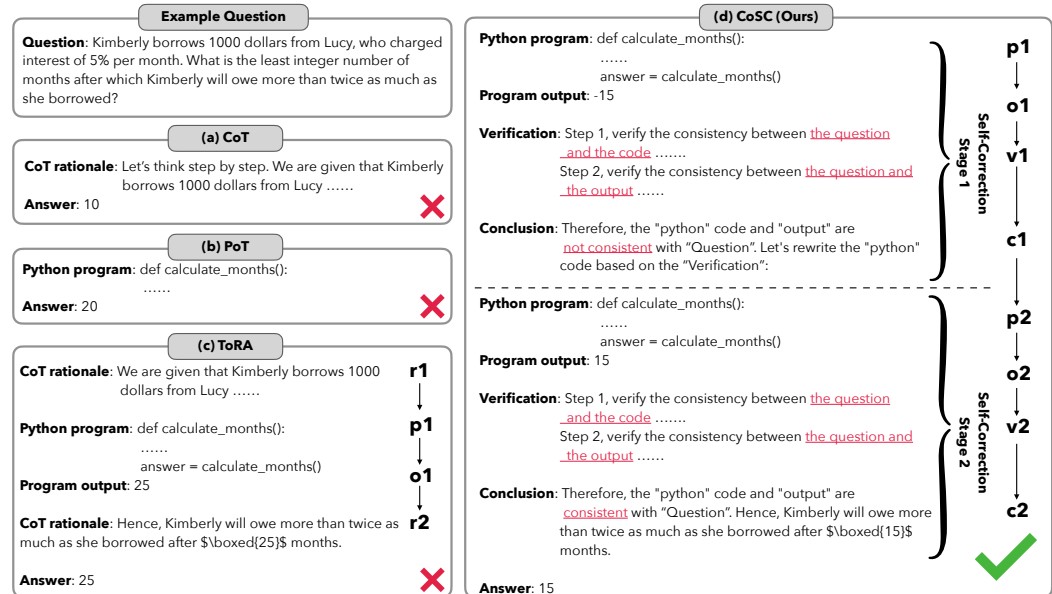

Figure 1: Comparison of four reasoning frameworks for solving an example mathematical question. (a) Chain of Thoughts (CoT) (Wei et al., 2022). (b) Program of Thoughts (PoT) (Chen et al., 2023a). (c) ToRA (Gou et al., 2023b) that incorporates CoT, PoT, and the utilization of tools. (d) Our proposed CoSC consists of a sequence of multiple self-correction stages (two stages are shown in this example). Each stage has four sub-stages: (**p1**) LLMs generate program *w.r.t.* the question; (**o1**) execute the program to obtain program output; (**v1**) perform two-step verification for consistency of the question with both the generated program and the program output; (**c1**) conclude a refined answer or continue the next subsequent self-correction stage depending on the verification result. The final answer is extracted from the last conclusion sub-stage with regular expression matching.

has become a favored technique among them, which updates open-sourced LLMs based on previously generated high-quality question-response pair datasets. Compared to open-source LLMs, finetuning has demonstrated significant improvement, but there is still potential room for further enhancement.

Most current methods (Yu et al., 2023; Gou et al., 2023b) generate finetuning datasets by prompting GPT-4 to rephrase mathematical questions from various perspectives or incorporate chain-of-thoughts (CoT) analysis (Wei et al., 2022) and program-of-thoughts (PoT) code (Chen et al., 2023a) to diversify mathematical responses. As a result, LLMs trained on these datasets can comprehend different questions and learn to use code to solve mathematical problems. However, precise multi-round reasoning capabilities remain challenging for them. Once potential errors occur throughout reasoning stages, it can lead to incorrect results. Consequently, it is essential to incorporate a self-correction mechanism into mathematical responses, which can enable LLMs to learn to correct themselves in multiple rounds. In addition, while supervised finetuning datasets can be developed with GPT-4's assistance, it still requires human experiments with different prompts and the cost of using the interface. This highlights the need for effective finetuning with unlabeled datasets.

To address the aforementioned challenges, our study introduces the Chain of Self-Correction (CoSC), a novel mechanism designed to embed self-correction as an inherent capability in LLMs, enabling them to validate and rectify their own results. The CoSC mechanism operates through a sequence of self-correction stages, where LLMs generate a program to solve a given problem, execute the program using program-based tools to obtain an output, and subsequently verify this output. Depending on the verification, the LLMs either advance to a subsequent stage of self-correction or conclude with the refined solution. An example of our CoSC reasoning trajectory with multiple self-correction stages (two stages are shown in this example) is shown in Fig. 1. This iterative self-correction process allows the LLMs to refine their reasoning steps and improve the accuracy of its mathematical reasoning.

To implement the CoSC mechanism at a low cost, we adopt a two-phase finetuning approach. In the first phase, termed the **CoSC foundational learning**, LLMs are trained with a relatively small volume of seeding data generated from GPT-4, equipping them with a baseline proficiency in the CoSC

methodology. In particular, we prompt GPT-4 with training questions from MATH (Hendrycks et al., 2021) and GSM8K (Cobbe et al., 2021) datasets to generate mathematical reasoning trajectories that adhere to the CoSC protocol. Specifically, each generated trajectory consists of program-of-thoughts code, program output, a two-step verification process that ensures the alignment of the question with both the generated program and the resulting output, and a conclusion to determine whether the trajectory should be refined or if the final answer can be provided.

Utilizing GPT-4 can be expensive, especially when dealing with large volumes of training data. Alternatively, we propose a cost-free method to further boost performance through self-enhancement in the second phase. This second phase, referred to as **CoSC self-enhancement**, builds upon the foundational learning by further adapting the LLMs obtained from the first phase with self-generated trajectories. These trajectories are produced by the models trained in the foundational phase, allowing for the generation of a substantial volume of data without additional GPT-4 intervention. In both phases, we retain only the trajectories whose answers match the ground-truth labels of the corresponding questions.

In summary, our study makes the following contributions:

- We propose the Chain of Self-Correction (CoSC) mechanism, which effectively embeds self-correction as an inherent ability in Large Language Models (LLMs). Once LLMs learn this ability during training, they can self-correct in a zero-shot setting during inference without the need for external feedback or few-shot demonstrations. With the inherent self-correction ability, even an originally weak LLM is able to achieve excellent performance in mathematical reasoning, as strongly supported by our experimental results. This unique contribution distinguishes our work from related works discussed in Section 2.2 of this field.

- To implement the CoSC mechanism at a low cost, we introduce a two-phase finetuning approach. The first phase involves CoSC foundational learning, where we use a relatively small volume of seeding data generated by GPT-4. In the second phase, CoSC self-enhancement occurs using a larger volume of self-generated data with the model obtained from the first phase, without relying on paid GPT-4.

- Our comprehensive experiments demonstrate that the CoSC mechanism provides a new benchmark for performance on established mathematical datasets when compared to existing open-source LLMs. Notably, our CoSC-Code-34B model achieves superior performance over both closed-source non-multi-modal LLMs and some of multi-modal LLMs, particularly on the challenging MATH dataset. These findings underscore the potential of our CoSC to improve the accuracy and reliability of LLMs in complex problem-solving domains.

- The proposed CoSC mechanism, by embedding self-correction as an inherent capability in LLMs, enables them to think before responding to a question, creating an internal chain of self-correction to progressively verify and rectify their original answers. It is more akin to the slow thinking process of humans, which is particularly helpful in solving difficult mathematical reasoning problems. This approach can provide valuable insights for future research and contribute to the ongoing advancement of LLMs across various domains.

## 2 RELATED WORK

### 2.1 LLMS FOR MATHEMATICAL REASONING

Mathematical reasoning (Liu et al., 2023a; Wang et al., 2023a; Huang et al., 2024b; Toshniwal et al., 2024; Chen et al., 2024a; Zhang et al., 2024) is a challenging reasoning task for LLMs, which requires the ability to understand mathematical concepts, computation and multi-round reasoning. Existing mathematical reasoning approaches can be broadly classified into three categories: (1) *Prompting* methods (Wei et al., 2022; Chen et al., 2023a; Wang et al., 2023b) focus on extracting the inherent mathematical reasoning skills of LLMs by utilizing well-crafted prompting strategies during inference. Notably, they leverage the existing knowledge in LLMs without the need for parameter updates. (2) *Pretraining* methods (Azerbayev et al., 2023; Fu et al., 2023; Yang et al., 2024) pre-train LLMs on large-scale corpora containing mathematical problems and related content with language modeling objectives. The goal is to train a base foundation language model as a platform for mathematical domain. (3) *Finetuning* methods (Luo et al., 2023; Yu et al., 2023; Yue et al., 2023;

Gou et al., 2023b; Liu et al., 2023b; Gao et al., 2024; Mitra et al., 2024) refine the mathematical reasoning capabilities of LLMs by offering more targeted training, which is inspired by distillation techniques (Furlanello et al., 2018; Hsieh et al., 2023). Central to this approach is the generation of high-quality question-response pair datasets, which needs the assistance of complexity-based chain-of-thoughts prompting or tools-based augmentation. In this paper, we propose a Chain of Self-Correction (CoSC) along this direction that incorporates an iterative self-correction process into datasets generation.

## 2.2 EXISTING METHODS RELATED TO SELF-CORRECTION IN LLMS

Mathematical reasoning poses a significant challenge due to its demand for precise multi-round logical reasoning to solve problems. The potential for errors increases with each reasoning step, making it crucial for LLMs to have the ability to self-correct in order to produce accurate results. There are some recent studies (Chen et al., 2023b; Gou et al., 2023a; Lightman et al., 2023; Huang et al., 2024a; Chen et al., 2024b) attempt to enable large language models to perform self-correction by either prompting methods or fine-tuning methods. For prompting methods, they can correct their responses when interacting with external tools (Gou et al., 2023a; Chen et al., 2023b), such as search engines and calculators, or designing complex prompts (Chen et al., 2024b). Notably, prompting methods can be orthogonally combined with finetuning methods. For finetuning methods, previous works (Yu et al., 2024; An et al., 2023) only model single-round correction during the training stage, and perform verification in a straightforward manner during the inference stage.

## 2.3 DIFFERENCE BETWEEN THE PROPOSED METHOD AND EXISTING SELF-CORRECTION TECHNIQUES

The existing self-correction research can be broadly classified into two categories: prompt-based methods (Gou et al., 2023a; Chen et al., 2023b; 2024b) and SFT-based methods (Yu et al., 2024; An et al., 2023). Our approach falls under the SFT-based category and fundamentally differs from prompt-based methods. Specifically, prompt-based methods leverage the intrinsic capabilities of large language models (LLMs) for mathematical reasoning. In contrast, our method embeds the chain of self-correction, a strong reasoning mechanism, as an inherent capability in the LLMs through parameter fine-tuning. Furthermore, compared to existing SFT-based methods, our approach models mathematical reasoning as a multi-round procedure during the training stage. In addition, these SFT-based works (Yu et al., 2024; An et al., 2023) perform the verification process in a straightforward manner, whereas our work conducts verification in a step-by-step manner by generating intermediate verification steps. Similar to the essence of the CoT approach, our model first verifies whether the generated code aligns with the problem description and then verifies whether the obtained result is consistent with the problem description. This step-by-step verification strategy significantly improves the verification process in self-correction. The excellent performance of our method on the challenging mathematical problem tasks, such as MATH and GSM8k, clearly demonstrates its effectiveness compared to the existing SFT-based methods (Yu et al., 2024; An et al., 2023).

## 3 METHOD

### 3.1 OVERVIEW

We propose a Chain of Self-Correction (CoSC) to address mathematical questions in a self-correction manner. Given a mathematical question denoted as $q$, the CoSC mechanism initiates the process by generating a program $p$ that comprises a Python function. The program $p$ is then executed using a Python interpreter to produce an output $o$. However, the program $p$ may contain bugs or incorrect intermediate code due to potential misinterpretations of the question $q$, leading to erroneous outputs. Unlike existing methods that either base their reasoning on incorrect intermediate content or halt reasoning upon encountering an error, the CoSC mechanism introduces a self-correction rationale during the reasoning process. After the generation of program $p$ and its output $o$, the CoSC model analyzes them and generates a verification $v$. This verification includes suggestions for improvements to ensure the consistency of the program $p$ and its output $o$ with the mathematical question $q$, respectively. Following this, the CoSC mechanism draws a conclusion $c$ based on these suggestions, which is used to either refine the program $p$ or generate the final answer. This process is

Figure 2: The training of Chain of Self-Correction (CoSC) consists of two phases. The first phase, **(a) CoSC Foundational Learning**, trains LLMs with seeding data generated from proprietary models, equipping them with a baseline proficiency in the CoSC methodology. In particular, we prompt GPT-4 with training questions from MATH (Hendrycks et al., 2021) and GSM8K (Cobbe et al., 2021) datasets to generate mathematical reasoning trajectories that adhere to the CoSC protocol. The second phase, **(b) CoSC Self Enhancement**, further adapts the seed model obtained from the previous phase with self-generated trajectories. These trajectories are produced by the seed model trained in the foundational phase, thereby enabling the generation of a substantial volume of data without the need for additional GPT-4 intervention. In both phases, we only retain trajectories whose answers match the ground-truth label.

repeated until a conclusive answer or a maximum limit of self-correction stages is reached, which can be summarized as $answer = povcpovc\cdots povc$. By employing this reasoning method, we can annotate public mathematical datasets and use the annotated multi-round, self-correction data to fine-tune LLMs.

## 3.2 TRAINING

To enable the CoSC mechanism, we propose a two-phase finetuning method. (1) **CoSC foundational learning** with seeding data generated from proprietary models. (2) **CoSC self enhancement** with self-generated data using the seed model obtained in the first training phase. A summary for training our CoSC is shown in Fig. 2.

### 3.2.1 CoSC FOUNDATIONAL LEARNING

Existing mathematical reasoning datasets, such as chain-of-thoughts (CoT) (Wei et al., 2022) and program-of-thoughts (PoT) (Chen et al., 2023a), primarily contain single-round annotations without multi-round, self-correction solutions for the reasoning process. This makes it challenging to fine-tune models to inherently possess self-correction capabilities. To address this issue, we utilize GPT-4 ($\mathcal{G}$) and a few-shot approach to annotate questions from publicly available mathematical datasets, including MATH (Hendrycks et al., 2021) and GSM8K (Cobbe et al., 2021), to generate seeding data with our Chain of Self-Correction (CoSC) mechanism. Table 1 compares our seeding data of CoSC with recently proposed mathematical reasoning datasets. The algorithm of the generation with our CoSC for each question from the training set is shown in Algorithm 1. Detailed instructions and example cases of our CoSC can be found in Appendix A and Appendix B.

Given a mathematical question $q$, the few-shot prompt used to call GPT-4 is defined as $\wp$. The corresponding trajectory, denoted as $\tau$, is generated through the following process. Firstly, we feed both the few-shot prompt $\wp$ and the question $q$ into GPT-4, which then generates a Python function code $p$ to solve the question $q$. It can be represented as:

$$\wp \oplus q \oplus \tau_{i-1} \rightarrow p_i, \tag{1}$$

where the symbol $\oplus$ represents concatenation and $\tau_i$ indicates the trajectory in the $i$ interaction round. After obtaining the code $p_i$, we execute it using a Python interpreter to acquire the runtime result

Table 1: The comparison to the previous mathematical reasoning datasets. Note that our CoSC only employs questions from the original training set of MATH and GSM8K datasets. Particularly, we provide the number of annotation data as #Annotation in each mathematical reasoning method. Additionally, Correction signifies whether the self-correction is to be utilized.

| Methods | #Annotation | Tool | Correction | LLM Used | Source |
|---|---|---|---|---|---|
| RFT (Yuan et al., 2023) | >100k | ✗ | ✗ | LLaMA-2 | GSM8K |
| Open-Platypus (Lee et al., 2023) | 25k | ✗ | ✗ | GPT-4 | 11 datasets with MATH |
| WizardMath (Luo et al., 2023) | >96k | ✗ | ✗ | ChatGPT | MATH & GSM8K |
| Lila (Mishra et al., 2022) | 134k | ✓ | ✗ | - | 20 datasets with MATH & GSM8K |
| MathInstruct (Yue et al., 2023) | 260k | ✓ | ✗ | GPT-4 | 14 datasets with MATH & GSM8K |
| ToRA (Gou et al., 2023b) | 16k | ✓ | ✗ | GPT-4 | MATH & GSM8K |
| CoSC (Ours) | 37k | ✓ | ✓ | GPT-4 | MATH & GSM8K |

$o_i$. Then, we enclose the result $o$ within "```output" and feed it, along with the previous few-shot prompt $\wp$, question $q$, the previous trajectory $\tau_i$, and generated code $p_i$ to the GPT-4. This produces a verification $v_i$ that analyzes the consistency between the question $q$ and the program code $p_i$, as well as between the question $q$ and the program output $o_i$. Additionally, it produces a conclusion $c_i$ on whether to proceed with the next round of reasoning to perform adjustment, denoted as:

$$\wp \oplus q \oplus \tau_{i-1} \oplus p_i \oplus o_i \rightarrow v_i \oplus c_i. \tag{2}$$

Depending on the conclusion $c_i$, it decides whether to terminate the reasoning process or proceed to the next round. If there is no inconsistency found during the verification step, it generates a natural language conclusion $c_i$ and encloses the answer to the problem with the special symbol "boxed{}". Otherwise, it should further revise the generated code $p_i$ in a new reasoning round by updating the trajectory $\tau_i$, denoted as:

$$\tau_i = \tau_{i-1} \oplus p_i \oplus o_i \oplus v_i \oplus c_i. \tag{3}$$

Based on the above steps, CoSC iteratively generates PoT code, program output, verification, and conclusion. At final, the reasoning process is stopped by checking whether the generated result contains an answer enclosed in "boxed{}".

To ensure that the self-correction can concisely and clearly analyze the code and its runtime results, we design the self-correction in a step-by-step format as follows:

- The first step involves verifying whether the generated program code $p$ is consistent with the question $q$, such as checking the variables and their relationships.
- The second step involves verifying whether the code runtime results $o$ meet the requirements of the question $q$, such as checking the reasonableness of numerical values.

Finally, for summary, CoSC will generate a conclusion $c$ to determine whether we should start the next round of reasoning or provide the final answer.

With CoSC, we use GPT-4 to annotate the MATH and GSM8K training datasets. To explore diverse data, we apply nucleus sampling (Holtzman et al., 2019) during GPT-4 annotation. Each question is sampled in 3 times. However, for some complex questions, we are unable to obtain even a single correct solution. For these problems, we apply 10 more samplings and retained up to 4 correct data. Finally, we filtered out incorrect answers and constructed 37k pieces of data using GPT-4.

Based on the 37k data constructed by GPT-4 as the seeding data, we apply CoSC foundational learning to train models. For a given question $q$, the response trajectory $\tau$ generated by GPT-4, denoted as $\tau = povcpovc \cdots povc$, the minimized negative log-likelihood loss used for training the model can be represented as:

$$arg \min_\theta \sum_{q,\tau} \sum_{i=1}^{n_q-1} -\log \mathbb{P}_\theta(p_{i+1}o_{i+1}v_{i+1}c_{i+1}|q, p_i \cdots o_i v_i c_i), \tag{4}$$

where $n_q$ is the iteration rounds of the question $q$ in our CoSC for the mathematical reasoning.

### 3.2.2 CoSC Self Enhancement

After the completion of the CoSC foundational learning, the seed model gains the ability to self-correct during inference and perform multi-round reasoning. Subsequently, we employ the seed

---

**Algorithm 1** Inference Reasoning with our CoSC

---

**Require:** question $q$, model $\mathcal{G}$, prompt $\wp$, external tools $\mathcal{E}$, stop condition $Stop(\cdot)$, iteration rounds $n_q$

1: $\tau_0 \leftarrow$ ""        ▷ Trajectory Initialization
2: **for** $i \leftarrow 1$ to $n_q$ **do**
3:     $p_i \sim \mathbb{P}_{\mathcal{G}}(\cdot|\wp \oplus q \oplus \tau_{i-1})$        ▷ Program Generation (Eq. 1)
4:     $o_i \leftarrow \mathcal{E}(p_i)$        ▷ Tool Execution
5:     $v_i \oplus c_i \sim \mathbb{P}_{\mathcal{G}}(\cdot|\wp \oplus q \oplus \tau_{i-1} \oplus p_i \oplus o_i)$     ▷ Verification and Conclusion Generation (Eq. 2)
6:     $\tau_i \leftarrow \tau_{i-1} \oplus p_i \oplus o_i \oplus v_i \oplus c_i$        ▷ Trajectory Adjustment (Eq. 3)
7:     **if** $Stop(c_i)$ **then**        ▷ Stopping Criteria
8:        **return** $\tau_i$
9:     **end if**
10: **end for**
11: **return** $\tau_n$

---

model, after CoSC foundational learning, to apply dense solution sampling and dense question sampling, which enables the generation of more self-generated data with self-correction mechanisms.

**Dense solution sampling**. For mathematical questions, there are usually multiple solutions, but using GPT-4 for annotating them is expensive. Therefore, we use the seed model after CoSC foundational learning to resample the questions in the datasets multiple times, which further improves the model generalization ability in answering questions. Specifically, we use the CodeLLaMA-34B model which has the best performance after CoSC foundational learning to perform dense nucleus-sampling on 16k training data questions. Each question in the MATH and GSM8K training datasets is sampled 64 times. We filter out the correct answers based on whether they match the ground-truth.

**Dense question sampling**. Similarly, there are multiple ways to ask a mathematical question. To improve the model generalization ability of question understanding, we use the data from Meta-Math (Yu et al., 2023) to generalize the questions. Each question in MetaMATH is rewritten by simply rephrasing it, self-verifying the conditions in the question, adding if-then questions to reverse the question conditions, *etc*. Specifically, we use the CodeLLaMA-34B model with CoSC foundational learning to perform nucleus-sampling on the questions, sampling each question once, and filtering out data points based on whether they match the ground-truth.

In the end, we obtain a total of 339k data points, including 37k seeding data generated from GPT-4 and 302k generated from the CodeLLaMA-34B model with CoSC foundational learning. Then, we adopt them to train models from scratch to obtain our final CoSC model.

### 3.3 IMPLEMENTATION DETAILS

By using 339k data points, including 37k seeding data and 302k self-generated data, we fine-tune the base models of LLaMA-2 (Touvron et al., 2023) and CodeLLaMA (Roziere et al., 2023) to obtain our CoSC and CoSC-Code, respectively. They have different parameter sizes, such as 7B, 13B, and 34B. All models use full-scale fine-tuning. We use the AdamW optimizer with a learning rate of 2e-5 for all models, with a batch size set to 128, training for 1 epoch. To enable training, we use DeepSpeed ZeRO stage 3 (Rajbhandari et al., 2021) and Flash-Attention 2 (Dao, 2023) to optimize the model's memory usage. During inference, we set a maximum of 3 calls to the Python interpreter and a maximum token length of 2048. The GPT-4 version for CoSC data generation is `gpt-4-0613`. Our experiments train the models in 7B size with 8 NVIDIA A800 80GB GPUs and train the models in 13B and 34B with 16 NVIDIA A800 80GB GPUs.

## 4 EXPERIMENTS

### 4.1 EVALUATION SETUP

**Datasets**. We evaluated models on the most widely used mathematical problem tasks, MATH (Hendrycks et al., 2021) and GSM8K (Cobbe et al., 2021). The MATH dataset encompasses a total of 12,500 problems, partitioned into 7,500 for training and 5,000 for testing. The GSM8K dataset

Table 2: Accuracy results (%) on MATH and GSM8K datasets. Vanilla models are tested with CoT. ZS indicates the zero-shot inference without demonstrations. PAL refers to the Program-Aided Language model prompting (Gao et al., 2023). The best results in each section are in **bold** and the second-best results are underlined. Our CoSC can outperform open-sourced LLMs, proprietary non-multi-modal LLMs and some of multi-modal LLMs, on the more challenging MATH dataset.

| Model | Base | Size | ZS | MATH | GSM8K | AVG |
|---|---|---|---|---|---|---|
| | | | | *Proprietary Models* | | |
| GPT-4o | - | - | ✗ | **76.6** (0-shot CoT) | 96.1 (8-shot CoT) | **86.4** |
| GPT-4V | - | - | ✗ | 52.9 (4-shot) | 92.0 (5-shot CoT) | 72.5 |
| GPT-4 (PAL) | - | - | ✗ | 51.8 (PAL) | 94.2 (PAL) | 73.0 |
| GPT-4 | - | - | ✗ | 42.5 (CoT) | 92.0 (5-shot CoT) | 67.3 |
| ChatGPT (PAL) | - | - | ✗ | 38.7 (PAL) | 78.6 (PAL) | 58.7 |
| ChatGPT | - | - | ✗ | 35.5 (CoT) | 80.8 (5-shot CoT) | 58.2 |
| Claude-3.5 Sonnet | - | - | ✗ | 71.1 (0-shot CoT) | **96.4** (0-shot CoT) | 83.8 |
| Claude-3 Opus | - | - | ✗ | 60.1 (0-shot CoT) | 95.0 (0-shot CoT) | 77.6 |
| Claude-2 | - | - | ✗ | 32.5 (0-shot CoT) | 85.2 (0-shot CoT) | 58.9 |
| Gemini-1.5 Pro | - | - | ✗ | 67.7 (4-shot Minerva) | 90.8 (11-shot) | 79.3 |
| Gemini-1.5 Flash | - | - | ✗ | 54.9 (4-shot Minerva) | 86.2 (11-shot) | 70.6 |
| Gemini-1.0 Ultra | - | - | ✗ | 53.2 (4-shot Minerva) | 88.9 (11-shot) | 71.1 |
| Gemini-1.0 Pro | - | - | ✗ | 32.6 (4-shot Minerva) | 77.9 (11-shot) | 55.3 |
| PaLM-2 | - | 540B | ✗ | 34.3 (4-shot CoT) | 80.7 (8-shot CoT) | 57.5 |
| | | | | *Open-Source Models* | | |
| LLaMA-2 | LLaMA-2 | 7B | ✗ | 4.1 (CoT) | 13.3 (CoT) | 8.7 |
| LLaMA-2 SFT | LLaMA-2 | 7B | ✓ | 7.2 | 41.3 | 24.3 |
| LLaMA-2 RFT | LLaMA-2 | 7B | ✓ | - | 51.2 | - |
| CodeLLaMA (PAL) | CodeLLaMA | 7B | ✗ | 16.6 (PAL) | 34.0 (PAL) | 25.3 |
| Platypus-2 (Lee et al., 2023) | LLaMA-2 | 7B | ✗ | 5.4 (CoT) | 14.4 (CoT) | 9.9 |
| WizardMath (Luo et al., 2023) | LLaMA-2 | 7B | ✓ | 10.7 | 54.9 | 32.8 |
| MetaMath (Yu et al., 2023) | LLaMA-2 | 7B | ✓ | 19.8 | 66.5 | 43.2 |
| ToRA (Gou et al., 2023b) | LLaMA-2 | 7B | ✓ | 40.1 | 68.8 | 54.5 |
| CoSC (Ours) | LLaMA-2 | 7B | ✓ | 42.7 | 70.5 | 56.6 |
| ToRA-Code (Gou et al., 2023b) | CodeLLaMA | 7B | ✓ | 44.6 | 72.6 | 58.6 |
| CoSC-Code (Ours) | CodeLLaMA | 7B | ✓ | **47.6** | **74.7** | **61.2 (+2.6)** |
| LLaMA-2 | LLaMA-2 | 13B | ✗ | 6.3 (CoT) | 24.3 (CoT) | 15.3 |
| LLaMA-2 SFT | LLaMA-2 | 13B | ✓ | 9.2 | 51.1 | 30.2 |
| LLaMA-2 RFT | LLaMA-2 | 13B | ✓ | - | 55.3 | - |
| CodeLLaMA (PAL) | CodeLLaMA | 13B | ✗ | 19.9 (PAL) | 39.9 (PAL) | 29.9 |
| Platypus-2 (Lee et al., 2023) | LLaMA-2 | 13B | ✗ | 7.1 (CoT) | 23.7 (CoT) | 15.4 |
| WizardMath (Luo et al., 2023) | LLaMA-2 | 13B | ✓ | 14.0 | 63.9 | 39.0 |
| MetaMath (Yu et al., 2023) | LLaMA-2 | 13B | ✓ | 22.4 | 72.3 | 47.4 |
| ToRA (Gou et al., 2023b) | LLaMA-2 | 13B | ✓ | 43.0 | 72.7 | 57.9 |
| CoSC (Ours) | LLaMA-2 | 13B | ✓ | 45.3 | 73.9 | 59.6 |
| ToRA-Code (Gou et al., 2023b) | CodeLLaMA | 13B | ✓ | 48.1 | 75.8 | 62.0 |
| CoSC-Code (Ours) | CodeLLaMA | 13B | ✓ | **50.3** | **77.3** | **63.8 (+1.8)** |
| LLaMA-1 RFT | LLaMA-1 | 34B | ✓ | - | 57.9 | - |
| CodeLLaMA (PAL) | CodeLLaMA | 34B | ✗ | 23.9 (PAL) | 53.3 (PAL) | 38.6 |
| ToRA-Code (Gou et al., 2023b) | CodeLLaMA | 34B | ✓ | 50.8 | 80.7 | 65.8 |
| CoSC-Code (Ours) | CodeLLaMA | 34B | ✓ | **53.5** | **82.3** | **67.9 (+2.1)** |

contains 8,500 problems, with 7,500 for training and 1,000 for testing. These datasets collectively encompass a broad range of mathematical questions, from basic arithmetic to competition level. More details of datasets are provided in Appendix C.1.

**Baselines**. For proprietary models, we present results from an array of SoTA LLMs, such as OpenAI's ChatGPT (gpt-3.5-turbo), OpenAI's GPT-4, OpenAI's GPT-4V, OpenAI's GPT-4o, Google's PaLM-2, Google's Gemini-1.0 Pro, Google's Gemini-1.0 Ultra, Google's Gemini-1.5 Flash, Google's Gemini-1.5 Pro, Anthropic's Claude-2, Anthropic's Claude-3 Opus, and Anthropic's Claude-3.5 Sonnet. By default, we report CoT prompting results, and include PAL (Program-Aided Language model) prompting (Gao et al., 2023) results for selected models. For open-source models, base models comprise LLaMA-2 (Touvron et al., 2023) and CodeLLaMA (Roziere et al., 2023) with

CoT and PAL prompting (Gao et al., 2023). Supervised Fine-Tuning (SFT) employs CoT rationales from the original MATH and GSM8K dataset (15k samples) for fine-tuning. Rejection sampling Fine-Tuning (RFT) (Yuan et al., 2023) leverages multiple models to generate diverse reasoning paths for fine-tuning. WizardMath (Luo et al., 2023) augments data using ChatGPT, and conducts SFT and RLHF. Platypus-2 (Lee et al., 2023) is fine-tuned with Open-Platypus reasoning datasets. ToRA (Gou et al., 2023b) uses GPT-4 to generate tool-used trajectories and finetunes on it. The prompting methods for each evaluation are shown in Appendix C.2.

**Metric**. We report accuracies of predicted answers. For numerical values, we perform rounding, while for expressions, we employ the Python library *sympy* for parsing.

## 4.2 MAIN RESULTS

In our experiments, we observe several findings that demonstrate the effectiveness of our proposed CoSC model on mathematical datasets. We show the experimental results in Table 2. First and foremost, our proposed CoSC consistently outperforms previous state-of-the-art open-source LLMs across all scales. Specifically, our CoSC-Code can achieve an average improvement of 2.6%, 1.8%, and 2.1%, on 7B, 13B, and 34B size, respectively.

Moreover, to further emphasize the superiority of our CoSC, we conduct a comprehensive comparison against multiple proprietary models. The results reveal that our CoSC-Code-34B can outperform all the advanced proprietary non-multi-modal LLMs, such as ChatGPT and GPT-4, as well as most advanced proprietary multi-modal LLMs, such as GPT-4V, Gemini-1.0 Pro, and Gemini-1.0 Ultra on MATH dataset. It is important to note that, unlike these proprietary models, our CoSC performs the inference in a zero-shot manner without demonstrations.

## 4.3 ABLATION STUDY

In our ablation study, we conduct experiments on MATH dataset with the largest number of 5,000 test samples with broad spectrum of subjects and difficulty levels. For the consideration of computational efficiency, we choose the smaller models of CodeLLaMA with size 7B and 13B as the base models.

### 4.3.1 EFFECT OF EACH TRAINING PHASE

Our CoSC mechanism comprises two training phases: (1) CoSC foundational learning and (2) CoSC self-enhancement. In order to assess the individual contributions of each training phase, we evaluate multiple combinations of the proposed training phases, the results of which are shown in Table 3.

The experiments conducted on CodeLLaMA serve as the baseline for our study. Initially, we employ GPT-4 to generate 37k seeding data for CoSC foundational learning, which equips LLMs with initial self-correction capabilities. As a result, we observe a notable improvement in performance on the MATH dataset, with accuracy increasing from 16.6% to 42.3% for the 7B size and from 19.9% to 47.0% for the 13B size.

To further enhance the self-correction performance, we utilize the seed model obtained in the first phase to generate additional data without relying on GPT-4. This approach leads to further improvements in performance on the MATH dataset from 42.3% to 47.6% and 47.0% to 50.3%, for 7B and 13B respectively.

It is exciting to see that the originally weak LLMs, such as the CodeLLaMA 7B base model and the 13B base model, can significantly improve in mathematical reasoning by using the proposed CoSC method, as shown in Table 3. This demonstrates the effectiveness of the CoSC method, which is able to embed self-correction as an inherent capability in LLMs, leading to significant improvements in mathematical reasoning performance.

### 4.3.2 EFFECT OF MULTI-ROUND REASONING IN THE PROPOSED CoSC MECHANISM

Our CoSC mechanism integrates a series of self-correction stages to progressively verify and refine output of LLMs. In particular, the conclusion step in our CoSC relies on clues from the verification step to determine whether to proceed to the next round of reasoning or directly provide the final answer. Such iterative multi-round mechanism enables LLMs to self-correct their outputs and

Table 3: Accuracy results (%) on the MATH dataset for the CoSC foundational learning and CoSC self enhancement. The results show the superiority and necessity of both training phases.

| Chain of Self-Correction (CoSC) | | 7B | 13B |
|---|---|---|---|
| Foundational Learning | Self Enhancement | | |
| ✗ | ✗ | 16.6 | 19.9 |
| ✓ | ✗ | 42.3 | 47.0 |
| ✓ | ✓ | 47.6 | 50.3 |

Table 4: The distribution of reasoning rounds on MATH test dataset across three LLMs in 7B and 13B size. Our CoSC-Code can generate more rounds compared to other models in mathematical reasoning. #Round indicates the number of reasoning rounds during inference.

| Models | 7B | | | 13B | | |
|---|---|---|---|---|---|---|
| | #Round=1 | #Round=2 | #Round=3 | #Round=1 | #Round=2 | #Round=3 |
| CodeLLaMA | 100% | 0 | 0 | 100% | 0 | 0 |
| ToRA-Code | 99.9% | 0.1% | 0 | 100% | 0 | 0 |
| CoSC-Code (Ours) | 78.3% | 12.7% | 9.0% | 79.3% | 13.1% | 7.6% |

Table 5: Accuracy results (%) for the test samples on the MATH dataset of our CoSC-Code with single-round reasoning and multi-round reasoning during the inference stage.

| Models | 7B | 13B |
|---|---|---|
| CoSC-Code with single-round reasoning | 40.2 | 42.4 |
| CoSC-Code with multi-round reasoning (Ours) | 47.6 | 50.3 |

improve accuracy. This experiment quantitatively investigates effect of this multi-round mechanism. As shown in Table 4, our CoSC can effectively generate more rounds of mathematical reasoning during inference, confirming efficacy of our CoSC mechanism in enhancing the reasoning process.

In addition, we conduct a comparison between single-round reasoning and multi-round reasoning using our CoSC mechanism on the test samples from the MATH dataset. The single-round results are obtained by extracting the answer solely from the output of the first round, without any self-correction enabled. To illustrate the impact of multi-round reasoning, we present the comparative results in Table 5. The results clearly demonstrate the effectiveness of multi-round reasoning in rectifying errors and improving overall accuracy. With the 7B model, we observe an improvement in accuracy from 40.2% to 47.6%, while with the 13B model, accuracy increases from 42.4% to 50.3%. These findings highlight the significant benefits of employing multi-round reasoning within our CoSC mechanism. More ablation studies are shown in Appendix D.

## 5 CONCLUSION

In conclusion, our Chain of Self-Correction (CoSC) mechanism equips Large Language Models (LLMs) with the ability to autonomously validate and refine their outputs. This mechanism facilitates a sequence of self-correction stages that progressively refine the reasoning process, leading to enhanced accuracy in mathematical reasoning. Our two-phase finetuning approach, involving initial training with proprietary model-generated seeding data and subsequent training with self-generated data, effectively enables the implementation of CoSC at a low cost. Through extensive experiments, we have demonstrated the remarkable performance improvement that CoSC brings to traditional mathematical datasets when compared to existing open-source LLMs, even advanced close-sourced non-multi-modal LLMs, such as ChatGPT, GPT-4 and multi-modal LLMs like GPT-4V, Gemini-1.0 Pro, and Gemini-1.0 Ultra. We believe that our CoSC mechanism can provide valuable insights for future research and contribute to the ongoing advancement of LLMs across various domains.

**Ethics statement.** We experiment on two mathematical datasets, including GSM8K and MATH, both of which use MIT License code. The prompts used in these experiments are listed in Appendix A, and we want to emphasize that none of the prompts contain any words that discriminate against any individual or group. Furthermore, prompts would not negatively impact anyone's safety in this work.

**Reproducibility statement.** We provide part of the codes and some seeding data. We will provide the remaining codes and data of CoSC for reproducibility upon the acceptance of the paper.

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

# A  PROMPT FOR OUR COSC

We present specific instructions and example few-shot prompts of our CoSC for querying GPT-4 to generate the seeding data.

## A.1  PROMPT FOR OUR COSC ON MATH

The prompt for CoSC on MATH is as follows:

```
Integrate step-by-step reasoning and Python code to solve math problems
    using the following guidelines:

 - Analyze the question and write functions to solve the problem; the
    function should not take any arguments.

 - Present the final results in LaTeX using a "\boxed{}" without any
    units.

 - Utilize the "pi" symbol and "Rational" from Sympy for $\pi$ and
    fractions, and simplify all fractions and square roots without
    converting them to decimal values.

 - In "verification", verify the consistency between the "Question" and
    the "python" code, as well as between the "Question" and "output" in
     "Solution"  step-by-step.

 - According to the "verification", rewrite the code if neccessary.

Here are some examples you may refer to:

---

Question: Find the matrix that corresponds to projecting onto the vector
    $\begin{pmatrix} 2 \\ -3 \end{pmatrix}.$

Solution:

```python
from sympy import Matrix, simplify

def projection_matrix():
    v = Matrix([2, -3])

    # Calculate the projection matrix
    P = (v * v.T) / (v.T * v)[0]

    # Simplify the elements
    P = P.applyfunc(simplify)

    return P

answer = projection_matrix()
print(answer)
```

```output
Matrix([[4/13, -6/13], [-6/13, 9/13]])
```

```verification
Step 1. Verify the consistency between the question and the code. The "
    python" code defines a function binary_sum_diff() that calculates the
     projection matrix for the given vector. The vector is defined as `v
    = Matrix([2, -3])`, and then use standard formula `(v * v.T) / (v.T *
```
```

```
        v)[0]` for calculating the projection matrix onto a vector, finally
    use `applyfunc(simplify)` to simplify the elements of the matrix.
Step 2: Verify the consistency between the question and the output. The
    answer matrix of 2 dimension vector should be a 2x2 matrix, the "
    output" of "python" code is `[[4/13, -6/13], [-6/13, 9/13]]`, which
    is reasonable.
```
```

```conclusion
Therefore, the "python" code and "output" are consistent with "Question".
    The matrix is $\boxed{\begin{pmatrix} \frac{4}{13} & -\frac{6}{13} \
    -\frac{6}{13} & \frac{9}{13} \end{pmatrix}}.$
```

---

Question: Compute $1011_2 + 101_2 - 1100_2 + 1101_2$.  Express your
    answer in base 2.

Solution:

```python
def binary_sum_diff():
    # Convert binary to decimal
    num1 = int("1011", 2)
    num2 = int("101", 2)
    num3 = int("1100", 2)
    num4 = int("1101", 2)

    # Perform the operations
    result = num1 + num2 - num3 + num4

    # Convert the result back to binary
    result_binary = format(result, "b")

    return result_binary

answer = binary_sum_diff()
print(answer)
```

```output
10001
```

```verification
Step 1. Verify the consistency between the question and the code. The "
    python" code defines a function binary_sum_diff() that convert the
    binary numbers to decimal, performing the operations, and then
    converting the result back to binary. It assigns the decimal form of
    values $1011, 101, 1100, 1101$ to $num1, num2, num3, num4$
    respectively, and then perform the operation $num1 + num2 - num3 +
    num4$, finally convert the result back to binary.
Step 2: Verify the consistency between the question and the output. The "
    output" of "python" code is $10001$, which is reasonable.
```

```conclusion
Therefore, the "python" code and "output" are consistent with "Question".
    The answer expressed in base 2 is $\boxed{10001_2}$.
```

---

Question: Convert the point $(0, -3 \sqrt{3}, 3)$ in rectangular
    coordinates to spherical coordinates.  Enter your answer in the form
    $(\rho,\theta,\phi),$ where $\rho > 0,$ $0 \le \theta < 2 \pi,$ and
    $0 \le \phi \le \pi.$

Solution:

```python
from sympy import sqrt, atan2, acos, pi

def rectangular_to_spherical():
    x, y, z = 0, -3*sqrt(3), 3
    rho = sqrt(x**2 + y**2 + z**2)
    theta = atan2(y, x)
    phi = acos(z/rho)
    return rho, theta, phi

answer = rectangular_to_spherical()
print(answer)
```

```output
(6, -pi/2, pi/3)
```

```verification
Step 1. Verify the consistency between the question and the code. The "
    python" code defines a function rectangular_to_spherical() that
    convert rectangular coordinates to spherical coordinates. It assigns
    the values $0, -3 \sqrt{3}, 3$ to $x, y, z$ respectively, and then
    use the following formulas $\rho = \sqrt{x^2 + y^2 + z^2}$, $\theta =
    \arctan{\frac{y}{x}}$ 3, $\phi = \arccos{\frac{z}{\rho}}$ to
    calculates $rho, theta, phi$.
Step 2: Verify the consistency between the question and the output. The
    required form of the "output" is $(\rho, \theta, \phi)$, where $\rho
    > 0$, $0 \le \theta < 2\pi$, and $0 \le \phi \le \pi$. The "output"
    of "python" code is $(6, -\pi/2, \pi/3)$, which is not reasonable.
```

```conclusion
Therefore, the "python" code and "output" are not consistent with "
    Question". Let's rewrite the "python" code based on the "verification
    ":
```

```python
from sympy import sqrt, atan2, acos, pi

def rectangular_to_spherical():
    x, y, z = 0, -3*sqrt(3), 3
    rho = sqrt(x**2 + y**2 + z**2)
    theta = atan2(y, x)
    phi = acos(z/rho)
    theta = (theta + 2 * pi) % (2 * pi)
    phi = (phi + pi) % pi
    return rho, theta, phi

answer = rectangular_to_spherical()
print(answer)
```

```output
(6, 3*pi/2, pi/3)
```

```
```verification
Step 1. Verify the consistency between the question and the code. The "
    python" code was rewritten, `theta = (theta + 2 * pi) % (2 * pi)` and
    `phi = (phi + pi) % pi` were added to meet requirement that $0 \le \
    theta < 2\pi$, and $0 \le \phi < \pi$.
Step 2: Verify the consistency between the question and the output. The
    required form of the "output" is $(\rho, \theta, \phi)$, where $\rho
    > 0$, $0 \le \theta < 2\pi$, and $0 \le \phi \le \pi$. The "output"
    of "python" code is $(6, 3*\pi/2, \pi/3)$, which is reasonable.
```

```conclusion
Therefore, the "python" code and "output" are consistent with "Question".
    Hence, the spherical coordinates for the point $(0, -3\sqrt{3}, 3)$
    are $\boxed{(6, \frac{3\pi}{2}, \frac{\pi}{3})}$.
```

---

Question: Solve the inequality
\[\frac{x^2 - 25}{x + 5} < 0.\]

Solution:

```python
from sympy import symbols, simplify
from sympy.solvers.inequalities import solve_univariate_inequality
from sympy.core.relational import LessThan

def solve_inequality():
    x = symbols('x')
    expression = (x**2 - 25) / (x + 5)
    inequality = LessThan(expression, 0)
    solution = solve_univariate_inequality(inequality, x, relational=
        False)
    simplified_solution = simplify(solution)

    return simplified_solution

answer = solve_inequality()
print(answer)
```

```output
Union(Interval.open(-oo, -5), Interval.Lopen(-5, 5))
```

```verification
Step 1. Verify the consistency between the question and the code. The "
    python" code defines a function solve_inequality() that that solves
    the inequality \[\frac{x^2 - 25}{x + 5} < 0.\]. It defines a symbolic
     variable `x` and the expression is defined as `\[\frac{x^2 - 25}{x +
     5}\]`, then use function `LessThan` to define the inequality and
    function `solve_univariate_inequality` to solve the inequality for `x
    `, finally use `simplify` to simplify the result.
Step 2: Verify the consistency between the question and the output. The
    required answer `x` should meet the condition: $x + $5 != $0, the "
    output" of "python" code is `(-oo, -5)` and `(-5, 5)`, which is
    reasonable.
```

```conclusion
Therefore, the "python" code and "output" are consistent with "Question".
     The answer is $\boxed{(-\infty,-5)\cup(-5,5)}$
```
```

```
```

---

Question: In the figure, triangles $ABC$ and $BCD$ are equilateral
    triangles. What is the value of $AD \div BC$ when expressed in
    simplest radical form?

\begin{asy}
draw((0,0)--(5,8.7)--(10,0)--cycle);
draw((10,0)--(15,8.7)--(5,8.7));
label("$A$",(0,0),SW);
label("$B$",(5,8.7),N);
label("$C$",(10,0),SE);
label("$D$",(15,8.7),NE);
\end{asy}

Solution:

```python
from sympy import Rational, sqrt, simplify

def ad_divided_by_bc():

    x = 1  # Side length of equilateral triangles

    ad_squared = 2 * x**2 - 2 * x**2 * cos(2 * pi / 3)  # Using the law
        of cosines
    ad = sqrt(ad_squared)

    bc = x # BC is the side length of the equilateral triangles

    simplified_ratio = simplify(ad / bc)

    return simplified_ratio

answer = ad_divided_by_bc()
print(answer)
```

```output
sqrt(3)
```

```verification
Step 1. Verify the consistency between the question and the code. The "
    python" code defines a function ad_divided_by_bc() that calculates
    the ratio of the lengths of $AD$ and $BC$ in the given figure. The
    side length of the equilateral triangles `x` is set to 1, because $
    ABC$ and $BCD$ are equilateral triangles, the angle between $AB$ and
    $AD$ should be 2 * \pi / 3, then we use the law of cosines, the
    square length of $AD$ is thus `2 * x**2 - 2 * x**2 * cos(2 * pi / 3)
    `, and the length of $AD$ is the square root of that, finally use `
    simplify` to simplify the 'ad / bc'.
Step 2: Verify the consistency between the question and the output. The
    required answer $AD \div BC$ should be positive, the "output" of "
    python" code is $\sqrt{3}$, which is reasonable.
```

```conclusion
Therefore, the "python" code and "output" are consistent with "Question".
    The value of $AD \div BC$ is $\boxed{\sqrt{3}}$.
```

```
---
```

## A.2 PROMPT FOR OUR COSC ON GSM8K

The prompt for CoSC on GSM8K is as follows:

```
Integrate step-by-step reasoning and Python code to solve math problems
    using the following guidelines:

  - Analyze the question and write functions to solve the problem; the
      function should not take any arguments.

  - Present the final results in LaTeX using a "\boxed{}" without any
      units.

  - Utilize the "pi" symbol and "Rational" from Sympy for $\pi$ and
      fractions, and simplify all fractions and square roots without
      converting them to decimal values.

  - In "verification", verify the consistency between the "Question" and
      the "python" code, as well as between the "Question" and "output" in
       "Solution"  step-by-step.

  - According to the "verification", rewrite the code if neccessary.

Here are some examples you may refer to:

---

Question: Olivia has \$23. She bought five bagels for \$3 each. How much
    money does she have left?

Solution:

'''python
from sympy import simplify

def money_left():
    money_initial = 23
    bagels = 5
    bagel_cost = 3
    money_spent = bagels * bagel_cost
    remaining_money = simplify(money_initial - money_spent)
    return remaining_money

answer = money_left()
print(answer)
'''

'''output
8
'''

'''verification
Step 1. Verify the consistency between the question and the code. The "
    python" code defines a function "money_left()" that calculates the
    remaining money after Olivia buys the bagels. The initial money is
    set to 23, the number of bagels is set to 5, and the cost of each
    bagel is set to 3. The total cost of the bagels is calculated as
    bagels * bagel_cost, which is $5 * $3 = $15. The remaining money is
    calculated as money_initial - money_spent.
Step 2: Verify the consistency between the question and the output. The
    left money should be a positive number or zero, and the "output" of "
    python" code is $8 >= $0, which is reasonable.
```

```
```

```conclusion
Therefore, the "python" code and "output" are consistent with "Question".
    Olivia has $\boxed{8}$ dollars left.
```

---

Question: Michael had 58 golf balls. On tuesday, he lost 23 golf balls.
    On wednesday, he lost 2 more. How many golf balls did he have at the
    end of wednesday?

Solution:

```python
from sympy import simplify

def remaining_golf_balls():
    golf_balls_initial = 58
    golf_balls_lost_tuesday = 23
    golf_balls_lost_wednesday = 2
    golf_balls_left = golf_balls_initial - golf_balls_lost_tuesday - golf
        _balls_lost_wednesday
    remaining_golf_balls = simplify(golf_balls_left)
    return remaining_golf_balls

answer = remaining_golf_balls()
print(answer)
```

```output
33
```

```verification
Step 1. Verify the consistency between the question and the code. The "
    python" code defines a function remaining_golf_balls() that
    calculates the remaining golf balls. The initial number of golf balls
     is set to 58, the lost on Tuesday is set to 23 and the lost on
    Wednesday is set to 2. The remaining golf balls is calculated as golf
    _balls_initial - lost_on_Tuesday - lost_on_Wednesday.
Step 2: Verify the consistency between the question and the output. The
    left number of golf balls should be a positive number or zero, and
    the "output" of "python" code is $33 >= $0, which is reasonable.
```

```conclusion
Therefore, the "python" code and "output" are consistent with "Question".
    Michael had $\boxed{33}$ golf balls at the end of Wednesday.
```

---

Question: There were nine computers in the server room. Five more
    computers were installed each day, from monday to thursday. How many
    computers are now in the server room?

Solution:

```python
from sympy import simplify

def total_computers():
    computers_initial = 9
```

```
      computers_per_day = 5
      num_days = 3
      computers_added = computers_per_day * num_days
      computers_total = simplify(computers_initial + computers_added)
      return computers_total

answer = total_computers()
print(answer)
```

```output
24
```

```verification
Step 1. Verify the consistency between the question and the code. The "
    python" code defines a function total_computers() that calculates the
     number of computers now. The initial number of computers is set to
    9, the installed computer each day is set to 5, the number of days is
     set to 3, the number of computer now in the server room is
    calculated as computers_per_day * num_days + computers_initial,
    however there are 4 days from monday to thursday.
Step 2: Verify the consistency between the question and the output. The
    number of computers should be a positive number or zero, and the "
    output" of "python" code is $24 >= $0, which is reasonable.
```

```conclusion
Therefore, the "python" code is not consistent with "Question". Let's
    rewrite the "python" code based on the "verification":
```

```python
from sympy import simplify

def total_computers():
    computers_initial = 9
    computers_per_day = 5
    num_days = 4 # from monday to thursday
    computers_added = computers_per_day * num_days
    computers_total = simplify(computers_initial + computers_added)
    return computers_total

answer = total_computers()
print(answer)
```

```output
29
```

```verification
Step 1. Verify the consistency between the question and the code. The "
    python" code was rewritten, `num_days` is set to 4, which consistent
    with the `from monday to thursday` in "Question".
Step 2: Verify the consistency between the question and the output. The
    number of computers should be a positive number or zero, and the "
    output" of "python" code is $29 >= $0, which is reasonable.
```

```conclusion
Therefore, the "python" code are consistent with "Question". There're $\
    boxed{29}$ computers in the server room.
```

```
---
```
```

```

# B  CASE OF OUR COSC ON MATH AND GSM8K TESTING SETS

We would like to show examples how our CoSC model generate a corresponding answer when a question is queried.

## B.1  CASE ON MATH TESTING SET

The case for CoSC on MATH testing set is as follows, where the question is input into our CoSC model and the solution represents the corresponding answer generated from our CoSC model.

```
Question: Find the sum of all values of x such that \abs{x-1}=7.

Solution:

```python
from sympy import symbols, solve, Abs

def solve_absolute_equation():
    x = symbols('x')
    equation = Abs(x - 1) - 7
    solutions = solve(equation, x)
    sum_solutions = sum(solutions)
    return sum_solutions

answer = solve_absolute_equation()
print(answer)
```

```output
Program output: NotImplementedError: solving Abs(x - 1) when  the
    argument is not real or imaginary.
```

```verification
Step 1. Verify the consistency between the question and the code. The "
    python" code defines a function solve_absolute_equation() that solves
     the absolute equation $|x-1| = 7$. It defines a symbolic variable `x
    ` and the equation is defined as `Abs(x - 1) - 7`, then use function
    `solve` to solve the equation for `x`, finally use `sum` to get the
    sum of all solutions.
Step 2: Verify the consistency between the question and the output. The "
    output" of "python" code is `NotImplementedError: solving Abs(x - 1)
    when the argument is not real or imaginary.`, which is not reasonable
    .
```

```conclusion
Therefore, the "python" code and "output" are  not consistent with "
    Question". Let's rewrite the "python" code based on the "Verification
    ".
```

```python
from sympy import symbols, solve

def solve_absolute_equation():
    x = symbols('x')
    equation1 = x - 1 - 7
    equation2 = 1 - x - 7
```

```
      solutions1 = solve(equation1, x)
      solutions2 = solve(equation2, x)
      sum_solutions = sum(solutions1) + sum(solutions2)
      return sum_solutions

answer = solve_absolute_equation()
print(answer)
```

```output
2
```

```verification
Step 1. Verify the consistency between the question and the code. The "
    python" code was rewritten, `equation1 = x - 1 - 7` and `equation2 =
    1 - x - 7` were added to solve the two possible equations separately,
     and then get the sum of all solutions.
Step 2: Verify the consistency between the question and the output. The
    required answer is the sum of all solutions of the equation $|x-1| =
    7$, the "output" of "python" code is $2$, which is reasonable.
```

```conclusion
Therefore, the "python" code and "output" are consistent with "Question".
     The sum of all values of $x$ such that $|x-1| = 7$ is $\boxed{2}$.
```

## B.2 CASE ON GSM8K TESTING SET

The case for CoSC on GSM8K testing set is as follows, where the question is input into our CoSC model and the solution represents the corresponding answer generated from our CoSC model.

```
Question: A shop sells school supplies. One notebook is sold at \$1.50
    each, a pen at \$0.25 each, a calculator at \$12 each, and a geometry
     set at \$10. Daniel is an engineering student, and he wants to buy
    five notebooks, two pens, one calculator, and one geometry set. The
    shop gives a 10\% discount on all the purchased items. How much does
    Daniel have to spend on all the items he wants to buy? (\textcolor{
    green}{Ground truth is 27})

Solution:

```python
from sympy import simplify, Rational

def total_cost():
    notebook_cost = 1.5
    pen_cost = 0.25
    calculator_cost = 12
    geometry_set_cost = 10
    discount = Rational(10, 100)
    notebooks = 5
    pens = 2
    calculators = 1
    geometry_sets = 1
    total_cost_before_discount = notebook_cost * notebooks + pen_cost *
        pens + calculator_cost * calculators + geometry_set_cost *
        geometry_sets
    total_discount = total_cost_before_discount * discount
    total_cost_after_discount = simplify(total_cost_before_discount * (1
        - discount))
    return total_cost_after_discount
```

```
answer = total_cost()
print(answer)
```

```output
27.0000000000000
```

```verification
Step 1. Verify the consistency between the question and the code. The "
    python" code defines a function total_cost() that calculates the
    total cost after discount. The cost of each item and the discount are
     set to their respective values. The total cost before discount is
    calculated as the sum of the cost of each item multiplied by their
    quantities. The total discount is calculated as total_cost_before_
    discount * discount. The total cost after discount is calculated as
    total_cost_before_discount - total_cost_after_discount.
Step 2: Verify the consistency between the question and the output. The
    total cost after discount should be a positive number or zero, and
    the "output" of "python" code is $27 >= $0, which is reasonable.
```

```conclusion
Therefore, the "python" code is consistent with "Question". Daniel has to
    spend $\boxed{27}$ dollars on all the items he wants to buy.
```

## C  EVALUATION SETUP DETAILS

### C.1  DATASETS DETAILS

**MATH** (Hendrycks et al., 2021). This dataset consists of competition level mathematics problems. It encompasses a total of 12,500 problems, partitioned into 7,500 for training and 5,000 for testing. Each problem is accompanied by a step-by-step solution and concludes with a distinct final answer, which is formatted for straightforward verification of the model generated solutions. Notably, the MATH dataset spans a broad spectrum of subjects and difficulty levels, including seven categories: Prealgebra, Algebra, Number Theory, Counting and Probability, Geometry, Intermediate Algebra, and Precalculus.

**GSM8K** (Cobbe et al., 2021). Comprising a diverse collection of grade school mathematical word problems, GSM8K is recognized for its high quality. While it is generally considered less challenging than the MATH dataset, it similarly provides step-level solutions with basic arithmetic operations (addition, subtraction, multiplication, division). The GSM8K dataset contains 8,500 problems, with 7,500 for training and 1,000 for testing.

### C.2  PROMPTING METHODS FOR EACH EVALUATION

Table 2 includes an identifier "ZS", which denotes whether the LLMs are evaluated in a zero-shot inference setting without demonstrations. To clarify further, we summarize below the prompting methods employed for each evaluation.

Proprietary Models:

- GPT-4o (OpenAI, 2024): Zero-shot CoT prompting for MATH; 8-shot CoT prompting for GSM8K.
- GPT-4V (OpenAI, 2023b): 4-shot prompting for MATH; 5-shot CoT prompting for GSM8K.
- GPT-4 and ChatGPT (OpenAI, 2023a): CoT prompting for MATH; 5-shot CoT prompting for GSM8K.
- Gemini family (Team et al., 2024): 4-shot Minerva prompting for MATH; 11-shot prompting for GSM8K.

- Claude family (Anthropic, 2023): Zero-shot CoT prompting for both datasets.
- PaLM-2 (Anil et al., 2023): 4-shot CoT prompting for MATH; 8-shot CoT prompting for GSM8K.

Open-Source Models:

- LLaMA-2 (Touvron et al., 2023) and Platypus-2 (Lee et al., 2023): CoT prompting for both datasets.
- CodeLLaMA (Roziere et al., 2023): Program-Aided Language (PAL) (Gao et al., 2023) prompting for both datasets.
- LLaMA-2 SFT (Yuan et al., 2023), LLaMA-2 RFT (Yuan et al., 2023), WizardMath (Luo et al., 2023), MetaMath (Yu et al., 2023), ToRA (Gou et al., 2023b), and our CoSC method: Fully zero-shot, requiring no demonstrations.

## D    MORE ABLATION STUDIES

### D.1    ACCURACY OF THE VERIFICATION MODULE AND ERROR REDUCTION OF THE CORRECTION MODULE

Our CoSC framework comprises two main components: verification and correction. The verification component identifies potential erroneous reasoning, while the correction component generates improved reasoning to address the issues identified in the verification step. To evaluate the effectiveness of these components, we conducted ablation studies. We provide a detailed analysis of the benefits derived from both modules, reporting the accuracy of the verification module and the error reduction achieved by the correction module. The accuracy of the verification module reflects how precisely it identifies errors, whereas the error reduction of the correction module measures the rate at which errors are corrected from one round to the next. Specifically, it examines the proportion of solutions deemed incorrect in the i-th round that are successfully corrected in the (i + 1)-th round.

The results of CoSC-Code in 7B and 13B on MATH dataset are shown in Table 6. It can be observed that the verification of our CoSC-Code can iteratively refine its outputs with high accuracy, about 70%. Furthermore, for more difficult questions that require self-correction across multiple rounds, our CoSC-Code is still capable of successfully reducing errors by over 25%. This confirms the effectiveness of both the verification module and the correction module in our CoSC method.

### D.2    CONTEXT LENGTHS FOR DIFFERENT QUESTIONS

We explore the effect of the context lengths when addressing different questions over multiple rounds of self-corrections. We have calculated the percentage distribution of context lengths for CoSC-Code in 7B and 13B on MATH dataset, as shown in Table 7. These statistics indicate that all context lengths fall within the 4K range, which is well-suited for modern LLMs.

### D.3    ALLEVIATED ISSUES BY OUR COSC

Our CoSC primarily targets resolving two types of issues related to (a) code errors, such as NotImplementedError, and (b) inconsistency errors between outputs and the given questions. We have quantified the reduction in errors for the 7B models on MATH dataset, as demonstrated in the results below. It can be observed that these two types of errors are effectively minimized by our CoSC method. This reduction highlights the effectiveness of our CoSC approach in enhancing the accuracy and reliability of LLMs in mathematical reasoning tasks.

(a) Statistics for error reduction related to code errors: reduced from 931 to 167.

(b) Statistics for error reduction related to inconsistency errors: reduced from 310 to 1.

### D.4    RESULTS OF TORA UNDER COSC PROMPTS

To explore whether only using CoSC prompts can embed self-correction ability in LLMs, we conduct an experiment using the CoSC prompts for evaluation on ToRA-Code-7B. The CoSC prompts are

Table 6: The accuracy of the verification module and the error reduction rate achieved by the correction module of CoSC-Code in 7B and 13B on MATH dataset.

| Models | accuracy of the verification module | error reduction of the correction module |
|---|---|---|
| CoSC-Code-7B | 68.5% | 25.74% |
| CoSC-Code-13B | 70.4% | 26.25% |

Table 7: The context lengths when addressing different questions over multiple rounds of self-corrections for CoSC-Code in 7B and 13B on MATH dataset.

| Models | 0-1k | 1k-2k | 2k-3k | 3k-4k | >4k |
|---|---|---|---|---|---|
| CoSC-Code-7B | 84.88% | 13.68% | 1.28% | 0.16% | 0 |
| CoSC-Code-13B | 85.36% | 13.24% | 1.30% | 0.10% | 0 |

Table 8: Accuracy results (%) on MATH and GSM8K datasets of ToRA-Code-7B, ToRA-Code-7B under CoSC prompts and our CoSC-Code-7B.

| Models | MATH | GSM8K | AVG |
|---|---|---|---|
| ToRA-Code-7B | 44.6 | 72.6 | 58.6 |
| ToRA-Code-7B under CoSC prompts | 42.8 | 68.0 | 55.4 |
| CoSC-Code-7B (Ours) | 47.6 | 74.7 | 61.2 |

same as those used in CoSC seeding data generation in Appendix A. The results on the MATH and GSM8K datasets are shown in Table 8.

As shown in Table 8, applying CoSC prompting to ToRA not only fails to outperform the original ToRA model but also results in a decline in performance. As demonstrated in Table 4, ToRA inherently lacks the robust multi-round reasoning capabilities needed for effective self-correction. When CoSC prompting is applied, it introduces complexity that the model is ill-equipped to handle, leading to confusion and errors in the iterative process. Similarly, during the development of the CoSC algorithm, we also attempt to apply self-correction prompts to the base CodeLLaMA model. However, this approach did not yield good performance and was significantly lower than the previous state-of-the-art results in open-source models. This led us to adopt a fine-tuning strategy instead.

In contrast, our CoSC model, which integrates self-correction as an inherent capability via fine-tuning, achieves superior results on both datasets. These findings suggest that for open-source LLMs, few-shot prompting alone is insufficient to effectively enable self-correction. The lack of significant gains from prompting further underscores the limitations of relying solely on in-context examples. Therefore, we argue that embedding self-correction as an inherent capability through fine-tuning is essential for truly endowing LLMs with robust self-correction abilities.

Moreover, by integrating self-correction directly into the training process, our approach allows models to perform self-correction autonomously in a zero-shot setting during inference, eliminating the need for external feedback or few-shot demonstrations. This self-correction mechanism enables even weaker LLMs to achieve significant improvements in mathematical reasoning—enhancements that are unattainable through prompting methods alone. Additionally, our CoSC framework is open-source, making these advancements accessible to the broader research community. We believe this represents a pivotal step toward democratizing advanced reasoning capabilities and fostering further innovation.

## E  LIMITATIONS

In this paper, we propose a novel mechanism known as the Chain of Self-Correction (CoSC) designed to enhance the reasoning capabilities of Large Language Models (LLMs). While our research primarily concentrates on mathematical reasoning, we posit that the CoSC mechanism could be effectively utilized across a wider spectrum of applications to rectify errors produced by LLMs. However, due to constraints related to the length of this paper, a comprehensive exploration of this generalization will be reserved for future study.

