# OpenReview forum: "Embedding Self-Correction as an Inherent Ability in Large Language Models for Enhanced Mathematical Reasoning"
_ICLR.cc/2025/Conference — Submitted to ICLR 2025_

### Official Review · Reviewer_gRDy · 2024-10-24

**Soundness:** 3
**Presentation:** 3
**Contribution:** 2
**Rating:** 6
**Confidence:** 3

**Summary:**

This paper proposes a technique for improving mathematical reasoning with large language models which the authors call the  "Chain of Self-Correction (CoSC)". CoSC is designed to incorporate self-correction as an inherent ability in LLMs to iteratively refine their reasoning through multiple rounds of program generation, execution, and verification. The approach uses a two-phase fine-tuning process: first, training with a small dataset generated by GPT-4, and then self-enhancing with self-generated data, thereby reducing reliance on expensive models. On standard math reasoning datasets MATH and GSM8, the method outperforms a wide range of open source techniques and models that are tested in this work, and also performs better than some proprietary LLMs.

**Strengths:**

1. Interesting and cost-efficient two-phase fine-tuning approach. Using a two-phase approach (starting with seeding data from GPT-4 and then using self-generated data) is an interesting and effective way to reduce reliance on expensive models. The authors also include an ablation study that shows the significant improvements with both initial GPT4 and self-enhanced finetuning.
2. Evaluation is very broad and shows core value. The paper shows a very extensive evaluation with many proprietary and open source models as baselines, which is admirable. Their CoSC approach is also evaluated on three different sizes of LLMs up to 35B, where the approach consistently performs better than all other open source models on all sizes. Ablation studies also show the gains from multiple rounds of correction to show the value of the concept of chain of self correction, and further ablation studies are included in appendices.
3. Presentation and details. Well-structured and well-written paper. The problem is motivated well, and the technique, implementation and evaluation is clearly explained with details, and prompts and more details are provided in extensive appendices.

**Weaknesses:**

1. Novelty is not clear. In particular, I am not sure how exactly the CoSC technique is different from ToRA (Gou et al., 2023b). ToRA seems to also use the interleaved approach of program-generation, execution and rationales in multiple rounds (similar to the chain of correction you have here). One thing that seems different with CoSC is that you have explicit verification and conclusion steps and verification checks separately for question and output correctness. But those seem a bit like relatively smaller optimizations to guide the model to do more explicit reasoning. You also classify ToRA as under prompting approaches as opposed to fine-tuning in the related work discussion, but from reading their work it seems they are also fine tuning the models? (and in fact using GPT4 for initial data generation and also using two steps in training their models). But I am not completely sure of this - please clarify what exactly the significant differences in technique there are with ToRA.

2. Evaluation results seem not to show very drastic gains. The overall improvement over the best open source model (ToRA) is 1.8% - 2.6% for all sizes of models, which seems pretty incremental. On the other hand, the differences are pretty big with the best  proprietary models (e.g. GPT4o has 76.6% on MATH vs CoSC's 53.5%  and Claude 3.5 Sonnet has 96.4% on GSM8K vs CoSC's 82.3%), which indicates a much bigger scale of potential (attainable) improvement that could have been made. Also, I am curious if you can show the core value of the technique with a prompting-based approach over any proprietery model? You are already using prompts to train the model with GPT-4 - what if you used those same prompts to generate CoSC type trajectories on top of the best performing proprietary models like GPT4o and Sonnet - will it improve upon their results further? That will also help establish the core value of the technique apart from the fine-tuning gains you see over open source models.


Other comments:

- I'm not sure why it is stressed that CoSC can work with zero shot as opposed to proprietary models that require few shots - since it has been already fine tuned heavily on this task it is not surprising it does not need additional examples - whereas the proprietary models are very generic powerful LLMs so they need a few examples to orient them towards this specific form of reasoning tasks. So in a sense the heavy fine-tuning is already supposed to replace the few shot training right?

- This wording is pretty convoluted and confusing: "The results reveal that our CoSC-Code-34B can outperform all
the advanced proprietary LLMs, as well as most advanced proprietary multi-modal LLMs, such as GPT-4V, Gemini-1.0 Pro, and Gemini-1.0 Ultra". It can be misleading to say it outperforms "all the advanced proprietrary LLMs" as that covers ALL proprietary LLMs including the multimodal ones. It should be "all non-multi-modal proprietrary LLMs". (BTW, I am not sure why multi-modality distinction actually matters here?)

- Can you highlight in bold the best performing proprietary models in Table 2 (which I think is GPT4o and Claude 3.5 Sonnet).


Typos:
 - "There are some recent studies (Chen et al., 2023b; Gou et al., 2023a; Lightman et al., 2023; Huang
et al., 2024a; Chen et al., 2024b) attempt to enable large language models to perform self-correction
by either prompting methods or or fine-tuning methods."
- "rewrited" should be "rewrote" or "rewritten" in multiple places
- "to obtain the final our CoSC model."
- "CoSC consisits of"

**Questions:**

1. Please explain in detail what are the important differences in technique of your approach and ToRA.
2. Can you compare a prompt-only version of your approach on the best proprietary models and show improvements over them?

---

> ### Author Response · Authors · 2024-11-25
> **Rebuttal by Authors [1/2]**
>
> Thank you for your valuable review and suggestions. Below we respond to the comments in **Weaknesses (W)** and **Questions (Q)**.
>
> ---
>
> ***W1 & Q1: Novelty is not clear. Please explain in detail what are the important differences in technique of your approach and ToRA.***
>
> Thank you for your valuable comment. **We would like to clarify that ToRA is also fine-tuned on data generated by GPT-4,** rather than utilizing a prompting method, as mentioned in Line 86-87 and Line 437-438 of the original paper.
>
> Although both our method and ToRA are fine-tuning approaches, there are fundamental differences in their underlying principles and functionalities. ToRA integrates Chain-of-Thought (CoT) and Program-of-Thought (PoT) strategies to enhance reasoning capabilities in mathematical problem-solving. **However, it is primarily limited to single-round reasoning and lacks self-correction capabilities.** Notably, while the design of ToRA may imply multi-round reasoning, it does not explicitly support multi-round inference. The reasoning process in ToRA typically involves only a single round, with additional iterations occurring only in extremely rare cases, as demonstrated in Table 4, where the number of such iterations is nearly zero.
>
> **In contrast, our method incorporates self-correction as an intrinsic feature of LLMs and is explicitly designed to support multi-round reasoning.** This enables the model to iteratively refine its responses, correcting errors and improving accuracy over several round of reasoning. These advancements contribute to a more robust, iterative approach to problem-solving. Experimental results clearly show that our method outperforms ToRA by a clear margin across all three model sizes (7B, 13B, and 34B) on both the MATH and GSM8K mathematical benchmark datasets. This underscores the superior effectiveness of our approach in enhancing mathematical reasoning capabilities.
>
> ---
>
> ***W2 & Q2: Evaluation results seem not to show very drastic gains. Can you compare a prompt-only version of your approach on the best proprietary models and show improvements over them? Will it improve upon their results further if you use GPT-4o to generate training data?***
>
> Thank you for your insightful comments. Regarding the performance gap between our method and proprietary models such as GPT-4o and Claude 3.5 Sonnet, we acknowledge that there is notable room for improvement. **However, it is important to note that proprietary models operate on an entirely different scale in terms of training data volume, computational resources, and architectural design.** This makes direct comparisons inherently challenging. Our primary goal of this paper has been to advance the capabilities of open-source models using feasible computational resources, rather than directly competing with state-of-the-art proprietary systems.
>
> As suggested by the reviewer, we conducted an additional experiment comparing a prompt-only version of our CoSC approach on GPT-4o (one of the top proprietary models) with the original GPT-4o on the MATH and GSM8K datasets. The table below shows our results alongside the official GPT-4o benchmarks on these datasets. **Our results surpass the official GPT-4o performance on both datasets, suggesting that self-correction-style reasoning is a broadly effective strategy for enhancing the reasoning capabilities of LLMs.**
>
> |  | MATH   | GSM8k  |
> |-----------|------|------|
> | GPT-4o with CoSC prompt (Ours) | 77.0 | 97.1 |
> | GPT-4o [1,2]                      | 76.6 | 96.1 |
>
> [1] https://openai.com/index/hello-gpt-4o/. \
> [2] https://ai.meta.com/blog/meta-llama-3-1/.
>
> Finally, while CoSC prompts can enhance proprietary models, this approach relies on closed systems **that are inaccessible to the broader research community.** In contrast, **our CoSC method is an open-source framework,** providing self-correction capabilities that are publicly available. We believe this represents a critical step toward democratizing advanced reasoning capabilities and fostering further innovation within the research community.

---

> > ### Comment · Reviewer_gRDy · 2024-11-28
> >
> > Thank you to the authors their detailed response to my comments.
> >
> > I am still not clear on the significant differences in the technique with Tora. This statement is confusing:
> > "while the design of ToRA may imply multi-round reasoning, it does not explicitly support multi-round inference". They have a notion of "rationales" that are meant to check the results in each round and can make corrections that can go into the next round to refine the solution further right? Also, the fact that it does empirically go into multi-round reasoning (even if it may be less often than CoSC) shows that by design it does support it right? Isn't their support for multiple rounds designed to allow self-correction?  It may help to have a more detailed comparison table that clearly lists the concrete conceptual differences between CoSC and ToRA approaches (without any empirical differences).
> >
> > On the significance of the evaluation results, thank you very much for performing the additional evaluation to show the value of self-correction based solely on prompting. It is good that these show a consistent increase above the baseline, even though again the increase seems a bit incremental. Can you discuss the statistical significance of these 1-3% improvements (both over ToRA with your fine-tuned model and over proprietary models with your prompting-based approach).

---

> > > ### Author Response · Authors · 2024-11-29
> > > **Thank you for your feedback**
> > >
> > > Thank you for your feedback! Below we respond to the remaining questions.
> > >
> > > ---
> > >
> > > ***Q1: A more detailed comparison table that clearly lists the concrete conceptual differences between CoSC and ToRA approaches (without any empirical differences).***
> > >
> > > Thanks for your insightful comment. We provide a more detailed and clear comparison between ToRA and our CoSC as follows.
> > >
> > > The main goal of ToRA is to integrate **external tools** into natural language reasoning to enhance its reasoning capabilities. ToRA generates a sequence consisting of an initial rationale, a program, an output, and a second rationale for a given question. Specifically, the initial rationale is used to analyze the problem before generating the program, while the second rationale is used to generate the final answer after executing the code, as described in Fig. 2 of the ToRA paper [1]. As shown in Table 4 of our paper, we acknowledge that ToRA can perform multi-round reasoning, but this happens only in extremely rare instances. Based on our analysis of all multi-round cases in ToRA, it is important to emphasize that ToRA only regenerates rationales in a new round when **external execution failures** occur, such as runtime errors. Consequently, ToRA is unable to generate a result to be placed within the `\boxed{}` symbol, which serves as the stopping condition, and therefore proceeds to the next round, as stated in Lines 8-9 on Page 4 of the ToRA paper [1].
> > >
> > > The main goal of our CoSC is to teach LLMs using **inherent ability** to achieve self-correction. Different from ToRA, our CoSC generates a sequence consisting of a program, an output, a detailed verification, and a conclusion in one round for a given question. CoSC proceeds to the next round when errors are detected through self-correction. Specifically, our CoSC designs **a detailed two-step self-correction format** in Line 300-304 of our paper. It can teach LLMs how to perform self-correction by verifying the consistency among the question, the python program, and the program outputs. Our self-correction allows the model to autonomously identify and correct errors, akin to the slow thinking process of humans, offering greater flexibility even without external feedback.
> > >
> > > In short, CoSC leverages **inherent ability** for self-correction, reducing dependence on external tools, which enhances its autonomy and scalability. We highlight the differences between ToRA and our CoSC in the table below.
> > >
> > > | Difference | ToRA | CoSC (Ours) |
> > > |-----------|-----------|-----------|
> > > | **Principle** | integrating **external** tools into natural language reasoning | using **inherent ability** to achieve self-correction |
> > > | **When to correct** | only **external** execution failures, i.e., runtime errors | errors checked by **inherent ability**|
> > >
> > > [1] Zhibin Gou, Zhihong Shao, Yeyun Gong, Yujiu Yang, Minlie Huang, Nan Duan, Weizhu Chen, et al. Tora: A tool-integrated reasoning agent for mathematical problem solving. In ICLR, 2024.
> > >
> > > ---
> > >
> > > ***Q2: Can you discuss the statistical significance of these 1-3% improvements (both over ToRA with your fine-tuned model and over proprietary models with your prompting-based approach).***
> > >
> > > Thanks for your valuable feedback. Our experiments show that GPT-4o with our CoSC prompts performs slightly better than the original GPT-4o. On one hand, this demonstrates the broad effectiveness of the proposed method. On the other hand, the lack of a significant performance improvement reflects the limitations of the prompt-only version of our approach. This is one of the key reasons that we chose to focus on fine-tuning in this paper.
> > >
> > > In contrast to the prompting approach, our fine-tuning method on the CodeLLaMA base model results in a significant performance improvement. Specifically, our fine-tuning-based CoSC boosts the average accuracy on both datasets (MATH and GSM8K) by 35.9%, 33.9%, and 29.3% over CodeLLaMA for the 7B, 13B, and 34B model sizes, respectively.
> > >
> > > When comparing our CoSC model to ToRA, our method consistently outperforms ToRA across all three model sizes (7B, 13B, and 34B) on both the MATH and GSM8K datasets. This demonstrates the generalizability and stability of our approach. Specifically, on the MATH dataset—the more challenging benchmark—CoSC achieves improvements of 3%, 2.2%, and 2.7% in accuracy for the 7B, 13B, and 34B model sizes, respectively. These gains are particularly significant, considering the difficulty of the MATH dataset and the fact that ToRA is already considered a top-performing method among open-source models.

---

> > > > ### Comment · Reviewer_gRDy · 2024-12-01
> > > >
> > > > Thank you for the further details and providing the comparison table. The difference that ToRA only corrects when external tool errors are encountered is a good qualitative difference between the two approaches. However, overall, the notion of self-correction with respect to code and mathematical reasoning is becoming very prevalent in current research, so it is difficult to clearly separate the novelty of your technique. Perhaps you can also comment on the following related works in this area, which again use the LLM to critique and refine their solutions and also use code-based approaches in doing so, where I dont think correction is based only on runtime errors encountered (these would be relevant related works for you to discuss in the paper anyway I think).
> > > >
> > > > [1] Solving Challenging Math Word Problems Using GPT-4 Code Interpreter with Code-based Self-Verification. Zhou et al, ICLR 2024
> > > >
> > > > [2] Self-Refine: Iterative Refinement with Self-Feedback. Madaan et al, NeurIPS 2023.
> > > >
> > > > In terms of significance of the experimental results, you claim that the marginal difference is due to prompting rather than fine-tuning. I think to make this clasim it should be shown that prompting vs fine tuning on the same model would produce such a huge difference (e.g. the prompting approach on your open source models performs so much worse than your fine-tuned versions). However, the marginal improvements with respect to SoTA will still remain.
> > > >
> > > > Overall, my two concerns remain about a very distinct novelty in the technique and marginal empirical improvements over SoTA approaches. Hence I would prefer to keep my current rating of borderline accept.

---

> ### Author Response · Authors · 2024-11-25
> **Rebuttal by Authors [2/2]**
>
> ***W3: I'm not sure why it is stressed that CoSC can work with zero shot as opposed to proprietary models that require few shots.***
>
> Thanks for your careful reading and bringing this to our attention. Our emphasis on CoSC’s zero-shot capability was intended to highlight the practical usability of the model in real-world applications, where providing task-specific examples might not always be feasible.
>
> We agree that proprietary models, being more general-purpose and not fine-tuned for these specific tasks, require few-shot examples to adapt to this reasoning paradigm. This distinction underscores the trade-off between model generality and task-specific optimization.
>
> ---
>
> ***W4: It can be misleading to say it outperforms "all the advanced proprietrary LLMs" as that covers ALL proprietary LLMs including the multimodal ones. It should be "all non-multi-modal proprietrary LLMs". (BTW, I am not sure why multi-modality distinction actually matters here?)***
>
> Thank you for your thoughtful feedback. **We have revised the wording in our revised paper from "proprietary LLMs" to "non-multi-modal proprietary LLMs".**
>
> Regarding the distinction, while multi-modal LLMs process questions through language, **their visual capabilities can enhance understanding of spatial or geometric concepts, which may improve mathematical reasoning in tasks like Geometry.** Despite this, our CoSC models, trained solely on language, outperform some multi-modal models in specific mathematical benchmarks, demonstrating the effectiveness of our chain of self-correction approach in improving mathematical reasoning.
>
> ---
>
> ***W5: Can you highlight in bold the best performing proprietary models in Table 2 (which I think is GPT4o and Claude 3.5 Sonnet).***
>
> Thank you for your valuable suggestion. We have updated Table 2 to highlight the best-performing proprietary models in bold in our revised paper. For the MATH dataset, GPT-4o is the best-performing proprietary model, while for the GSM8K dataset, the top-performing proprietary model is Claude-3.5 Sonnet.
>
> ---
>
> ***W6: Typos.***
>
> Thank you for your careful review. We have corrected these typos in the revised version of our paper.

---

> ### Author Response · Authors · 2024-12-01
> **Author Response [1/2]**
>
> Thank you for your feedback! Below we respond to the follow-up questions.
>
> ---
>
> ***Q1: The notion of self-correction with respect to code and mathematical reasoning is becoming very prevalent in current research, so it is difficult to clearly separate the novelty of your technique.***
>
> Thank you for your insightful comment! We would like to highlight that the key differences between our CoSC and CSV [1] as well as Self-Refine [2] are outlined as follows.
>
> - **Difference from CSV [1].** The main difference between CSV [1] and our CoSC lies in the approach to verification. **CSV relies on external tools, specifically code, for verification, whereas our CoSC approach performs verification entirely in natural language without using tools,** which relies on the model's inherent capabilities for this process. CSV [1] argues in Table 4 of their original paper that relying solely on natural language verification of CSV can compromise accuracy and negatively impact performance. In contrast, our CoSC approach challenges this view and successfully achieves self-correction using only natural language verification, facilitated by a fine-tuning-based method. Specifically, our CoSC verifies the consistency between the question, the Python program, and the program's outputs using only natural language, which has proven effective.
>
> - **Difference from Self-Refine [2].** The primary difference between Self-Refine [2] and our CoSC is that **Self-Refine is unable to effectively identify mathematical errors, whereas our CoSC can do so.** As stated in Paragraph 4 of Section 3.3 in Page 5 of their original paper of Self-Refine [2], the modest performance improvements in mathematical reasoning stem from the inability of Self-Refine to accurately identify errors. Self-Refine cannot be applied effectively for weaker models, as stated in Page 7 in their paper. Despite leveraging powerful LLMs such as GPT-3, ChatGPT, and GPT-4, Self-Refine only achieves **a minimal accuracy improvement of 0%-0.2%** on mathematical reasoning tasks, as shown in Table 1 of their paper.  In contrast, our CoSC approach excels in error identification during mathematical reasoning. As demonstrated in Table 6 of our paper, the verification module in our CoSC achieves **a classification accuracy of approximately 70%, showcasing its ability to effectively identify erroneous answers.** Moreover, our CoSC is capable of significantly improving the performance of weaker LLMs compared to GPT-3, ChatGPT, or GPT-4 used in Self-Refine. Specifically, our fine-tuning-based CoSC yields significant improvements in mathematical reasoning, with average accuracy boosts of 35.9%, 33.9%, and 29.3% over CodeLLaMA on the MATH and GSM8K datasets for the 7B, 13B, and 34B model sizes, respectively.
>
> We acknowledge the valuable contributions of CSV [1] and Self-Refine [2]. Both approaches offer important insights, and we will ensure they are properly referenced and discussed in the next version of our paper. While our CoSC approach presents key distinctions, we believe that ongoing dialogue with these works is crucial for advancing the understanding and capabilities of self-correction methods in LLMs.
>
> [1] Solving Challenging Math Word Problems Using GPT-4 Code Interpreter with Code-based Self-Verification. In ICLR, 2024.\
> [2] Self-Refine: Iterative Refinement with Self-Feedback. In NeurIPS, 2023.

---

> ### Author Response · Authors · 2024-12-01
> **Author Response [2/2]**
>
> ***Q2: I think to make this clasim it should be shown that prompting vs fine tuning on the same model would produce such a huge difference (e.g. the prompting approach on your open source models performs so much worse than your fine-tuned versions). However, the marginal improvements with respect to SoTA will still remain.***
>
> Thank you for your valuable feedback. We conducted experiments comparing the performance of the prompt-based version and the fine-tuning version of our CoSC approach on the same base LLM, CodeLLaMA. The results are shown in the table below. We observed that the fine-tuning version of CoSC significantly outperforms the prompting version by 28.9% and 37.9% in accuracy on the MATH and GSM8K datasets, respectively. These results clearly demonstrate that fine-tuning is far more effective than prompting when embedding self-correction capabilities into LLMs.
>
> For context, Self-Refine [2] achieves only a 0%-0.2% accuracy improvement on the GSM8K dataset over the baseline, as shown in Table 1 of their original paper. CSV [1], which uses natural language-based self-verification without relying on code, even results in a 0.4% accuracy decrease on the MATH dataset compared to the baseline, as shown in Table 4 of the original CSV paper [1]. It is worth noting that the MATH dataset is a more challenging benchmark compared to GSM8K. However, our CoSC approach can achieve inherent self-correction capabilities without relying on any external tools and improve accuracy by 3%, 2.2%, and 2.7% over ToRA for the 7B, 13B, and 34B model sizes on the MATH dataset, respectively. Compared to the minimal or negligible improvements seen in CSV [1] and Self-Refine [2], we argue that the performance gains achieved by our CoSC approach are particularly significant, especially considering the challenging nature of the MATH dataset.
>
> |  | MATH | GSM8K |
> |-------|------|------|
> | CodeLLaMA-7B with CoSC prompt | 18.7 | 36.8 |
> | CodeLLaMA-7B with CoSC finetuning (Ours) | 47.6 | 74.7 |

---

### Official Review · Reviewer_KVMJ · 2024-11-01

**Soundness:** 2
**Presentation:** 2
**Contribution:** 2
**Rating:** 5
**Confidence:** 4

**Summary:**

Authors are studying whether an LLM that performs poorly on mathematical reasoning tasks can achieve a much stronger performance through a combination of fine-tuning and structured sampling with code execution. They find that their method produces very strong performance compared to proprietary and OS models and they analyze how their different interventions contribute to this performance increase.

**Strengths:**

The authors demonstrate that an initial seed model trained on GPT-4 generated data (distillation) can then be used to generate its own training data through self-correction (expert iteration), which
- reduces dependency on expensive API calls to GPT-4 after the initial seeding phase
- shows that a model can effectively act as its own teacher/critic through the Chain of Self-Correction mechanism
- demonstrates that relatively small amounts of high-quality seed data can go some distance towards bootstrap a more extensive self-improvement process
This methodology is powerful and the authors execute well on it.

**Weaknesses:**

The authors train a small model on the output of a large model and find that it improves performance a lot (compare "model distillation" - https://arxiv.org/abs/2305.02301) and then train the resulting model on its own output and find that it improves performance further (compare born again networks https://arxiv.org/abs/1805.04770 or more generally "expert iteration"). The authors should mention these highly related fields of research and contextualize their research better, especially a big step of performance increase comes from the distillation step.

The comparison landscape is incomplete and potentially misleading: Notable omission of open-source models with strong mathematical performance (QWEN2, Orca-Math, ...) that match/surpass the achieved performance.

The authors should indicate exactly which GPT-4 version was used for training data generation (presumably gpt-4o-2024-08-06 or gpt-4o-mini-2024-07-18?). This is not only important for reproducibility, but also for understanding how much of the gap between the weak model and GPT-4 the models were able to close via distillation.

The phrase "for some complex solutions we can only get < 1" is slightly awkward

**Questions:**

Do the experiments for Table 4 use a fewshot prompt for the non-CoSC models to tell them how to utilize multiple rounds of reasoning? How do the authors explain that the fraction of "more than one round of reasoning" very close to 0% for those models?

---

> ### Author Response · Authors · 2024-11-25
> **Rebuttal by Authors**
>
> Thank you for your valuable review and suggestions. Below we respond to the comments in **Weaknesses (W)** and **Questions (Q)**.
>
> ---
>
> ***W1: The authors train a small model on the output of a large model and find that it improves performance a lot (compare "model distillation" [1]) and then train the resulting model on its own output and find that it improves performance further (compare born again networks [2] or more generally "expert iteration").***
>
> Thank you for your insightful comment. We have now included references to these papers [1,2] on model distillation and expert iteration in Line 163 of our revised paper. Additionally, we have revised the paper to better contextualize our work and to explicitly acknowledge the contributions of these related research fields. Thank you for bringing this to our attention.
>
> [1] Cheng-Yu Hsieh, Chun-Liang Li, et al. Distilling Step-by-Step! Outperforming Larger Language Models with Less Training Data and Smaller Model Sizes. In ACL, 2023.\
> [2] Tommaso Furlanello, Zachary C. Lipton, et al. Born Again Neural Networks. In ICML, 2018.
>
> ---
>
> ***W2: The comparison landscape is incomplete and potentially misleading: Notable omission of open-source models with strong mathematical performance (QWEN2, Orca-Math, ...) that match/surpass the achieved performance.***
>
> Thank you for your valuable feedback. Qwen2 serves as a pretrained base LLM, whereas Orca-Math is fine-tuned on Mistral. In contrast, our CoSC model is fine-tuned on CodeLLaMA, highlighting the foundational differences in the base models. **These differences result in inherent disparities in model capabilities, which limit the direct comparison of the results in their current form.**
>
> Our primary objective in this work is not to achieve the best performance on mathematical dataset leaderboards but to evaluate the novel capabilities introduced by our method, such as self-correction and iterative reasoning. These features represent significant advancements in enhancing reasoning robustness and accuracy, which are independent of the choice of base model.
>
> **In this revised version, we have included a broader discussion of Qwen2 and Orca-Math in Section 2.1 to provide context for their performance.** However, given the limited time during the rebuttal phase, we plan to leverage our CoSC training dataset to fine-tune Qwen2 and Mistral in future work. This will enable a fairer, more direct, and comprehensive comparison of their reasoning capabilities under a unified framework.
>
> ---
>
> ***W3: The authors should indicate exactly which GPT-4 version was used for training data generation.***
>
> Thank you for your valuable suggestion. We have clarified that `gpt-4-0613` was used to generate our training data and added this information in Line 368 of the revised paper.
>
> ---
>
> ***W4: The phrase "for some complex solutions we can only get < 1" is slightly awkward.***
>
> Thanks for pointing that out. We have revised it accordingly in the updated version of the paper.
>
> ---
>
> ***Q1: Do the experiments for Table 4 use a fewshot prompt for the non-CoSC models to tell them how to utilize multiple rounds of reasoning? How do the authors explain that the fraction of "more than one round of reasoning" very close to 0% for those models?***
>
> Thank you for your insightful question. **We would like to clarify that no few-shot prompts were used for any of the models presented in Table 4.** The observation that the fraction of instances involving "more than one round of reasoning" is nearly 0% for the non-CoSC models, i.e., CodeLlama and ToRA-Code, highlights their limited capability to perform multi-round reasoning effectively. It is worth noting that both CodeLlama and ToRA-Code used in this work are the official open-source inference models provided by their respective developers. This result underscores the distinct advantage of CoSC in enabling multi-round reasoning processes.

---

> > ### Comment · Reviewer_KVMJ · 2024-11-26
> >
> > Thank you for your response, I am updating my score upwards. Similar to the other reviewers I remain unsure about how fair the comparison to existing methods is and whether it can justify e.g. the following in the abstract: "surpassing the performance of well-established models such as ChatGPT, GPT-4, and even multi-modal LLMs like GPT-4V, Gemini-1.0 Pro, and Gemini-1.0 Ultra."

---

> > > ### Author Response · Authors · 2024-11-28
> > > **Thank you for your feedback**
> > >
> > > Thank you for your feedback! Below we respond to the remaining questions.
> > >
> > > ---
> > >
> > > ***Q1: I remain unsure about how fair the comparison to existing methods is.***
> > >
> > > Thank you for your insightful comment. To address your concern about the fairness of the comparison, we conducted an experiment to directly compare our CoSC model with ToRA [1], the top-performing open-source method. For fairness, we used the same number of training samples (69K) as reported in the official ToRA paper, representing its best official result. The results, presented in the table below, clearly demonstrate that CoSC outperforms ToRA on both datasets.
> > >
> > > This result underscores that the superior performance of CoSC is not merely due to a larger training dataset but is instead attributed to the effectiveness of its self-correction mechanism. This mechanism enables CoSC to iteratively refine its outputs, providing a significant advantage over ToRA's approach. We believe this focused evaluation highlights the robustness and efficiency of our method under comparable settings.
> > >
> > > If you have additional suggestions or require further clarifications, we would be happy to address them.
> > >
> > > |  | MATH   | GSM8K  |
> > > |-----------|------|------|
> > > | ToRA-Code-7B [1] | 44.6 | 72.6 |
> > > | CoSC-Code-7B (Ours) | 47.0 | 74.2 |
> > >
> > > [1] Zhibin Gou, Zhihong Shao, Yeyun Gong, Yujiu Yang, Minlie Huang, Nan Duan, Weizhu Chen, et al. Tora: A tool-integrated reasoning agent for mathematical problem solving. In ICLR, 2024.
> > >
> > > ---
> > >
> > > ***Q2: I remain unsure about whether it can justify e.g. the following in the abstract: "surpassing the performance of well-established models such as ChatGPT, GPT-4, and even multi-modal LLMs like GPT-4V, Gemini-1.0 Pro, and Gemini-1.0 Ultra."***
> > >
> > > Thank you for your thoughtful feedback. As outlined in Lines 31-34 of our paper, our claim is specifically based on the performance of our CoSC-Code-34B model **on the MATH dataset,** where it achieves an accuracy of 53.5%. This surpasses the performance of several well-established models, including ChatGPT, GPT-4, and multi-modal models like GPT-4V, Gemini-1.0 Pro, and Gemini-1.0 Ultra, as reported in Table 2.
> > >
> > > To ensure fairness and validity, the results were obtained using the same evaluation protocol across all models, focusing on the MATH dataset—a well-recognized benchmark for mathematical reasoning tasks. By emphasizing this context, we aim to clarify that our claim is dataset-specific and grounded in empirical evidence, rather than a general statement about model superiority across all tasks or domains.
> > >
> > > We hope this clarification addresses your concerns and are happy to provide additional context or analysis if needed.

---

> > > > ### Author Response · Authors · 2024-12-02
> > > > **A Kind Reminder of Further Discussion**
> > > >
> > > > Dear Reviewer KVMJ,
> > > >
> > > > Thank you once again for your valuable comments and suggestions, which have been extremely helpful to us. We have posted detailed responses to the concerns you raised and have included additional experimental results.
> > > >
> > > > We fully understand that this is a particularly busy period, and **we deeply appreciate it if you could take some time to provide further feedback on whether our responses address your concerns.** If there are any additional comments, we will try our best to address them promptly.
> > > >
> > > > Sincerely,\
> > > > Authors of Submission 7154

---

### Official Review · Reviewer_VQBN · 2024-11-03

**Soundness:** 2
**Presentation:** 3
**Contribution:** 2
**Rating:** 5
**Confidence:** 3

**Summary:**

The paper describes a method called Chain of Self-Correction (CoSC). The idea is to generate a large set of synthetic data that includes stages of self-correction in order to fine-tune a model to learn self-correction capabilities. Then, at inference time, they employ this verification step to enhance performance on mathematical reasoning.

**Strengths:**

- The paper provides a framework for improving the self-correction abilities of models. They propose a framework to synthetically generate data for improving self correction.
- This synthetic data could be a useful resource for training models in the future.
- The performance seems good, and the method beats prior work fine-tuning on synthetically generated data.

**Weaknesses:**

- Compared to ToRA, the method only has about 3% improvement. However, while ToRA was only trained on 16k annotated samples, this method was trained on 37k, so the comparison is not apples to apples.
- There is no ablation to clarify what capabilities are gained at a more granular level. CoSC has a generation and a verification step, unlike previous methods. The paper does not analyze these separately. For example, what is the precision/recall of the verification step on the programs (how often does the verification step accidentally classify a correct program as wrong, or vice versa)?
- The main novelty of this paper is the verification/self-correction step. Without an extensive evaluation showing these capabilities have improved, it is hard to assess the effectiveness of the proposed method.

**Questions:**

- Did you try any normalizing experiment between CoSC and ToRA where both are trained on the same number of samples?
- Compared to ToRA and looking at Table 5, it seems like the accuracy when using just one round of reasoning is lower than ToRA. Does that mean the data generated is worse than ToRA's when it comes to reasoning without any self-correction steps?
- Because the models were trained on code, do they have an improved sense of code understanding?

---

> ### Author Response · Authors · 2024-11-25
> **Rebuttal by Authors [1/3]**
>
> Thank you for your valuable review and suggestions. Below we respond to the comments in **Weaknesses (W)** and **Questions (Q)**.
>
> ---
>
> ***W1 & Q1: The comparison is not apples to apples. Did you try any normalizing experiment between CoSC and ToRA where both are trained on the same number of samples?***
>
> Thank you for your insightful comment. Our method is not a continuation or follow-up to ToRA. Instead, it fundamentally differs from ToRA in both its underlying principles and functionality. ToRA integrates Chain-of-Thought (CoT) and Program-of-Thought (PoT) approaches to enhance the reasoning capabilities of LLMs in mathematical problem-solving. However, it is primarily restricted to single-round reasoning and lacks self-correction capabilities. In Table 4, we can see that the reasoning process in ToRA typically involves only a single round, with additional iterations occurring only in extremely rare cases. In contrast, our method incorporates self-correction as an intrinsic feature of LLMs and is explicitly designed to support multi-round reasoning. These advancements enable more robust and iterative problem-solving.
>
> To address your concern about the fairness of the comparison, we conducted an experiment to directly compare our CoSC model with ToRA [1], the top-performing open-source method. For fairness, we used the same number of training samples (69K) as reported in the official ToRA paper, representing its best official result. The results, presented in the table below, clearly demonstrate that CoSC outperforms ToRA on both datasets.
>
> This result underscores that the superior performance of CoSC is not merely due to a larger training dataset but is instead attributed to the effectiveness of its self-correction mechanism. This mechanism enables CoSC to iteratively refine its outputs, providing a significant advantage over ToRA's approach. We believe this focused evaluation highlights the robustness and efficiency of our method under comparable settings.
>
> |  | MATH   | GSM8K  |
> |-----------|------|------|
> | ToRA-Code-7B [1] | 44.6 | 72.6 |
> | CoSC-Code-7B (Ours) | 47.0 | 74.2 |
>
> [1] Zhibin Gou, Zhihong Shao, Yeyun Gong, Yujiu Yang, Minlie Huang, Nan Duan, Weizhu Chen, et al. Tora: A tool-integrated reasoning agent for mathematical problem solving. In ICLR, 2024.

---

> ### Author Response · Authors · 2024-11-25
> **Rebuttal by Authors [2/3]**
>
> ***W2: There is no ablation to clarify what capabilities are gained at a more granular level. CoSC has a generation and a verification step, unlike previous methods. The paper does not analyze these separately.***
>
> Thank you for your valuable comment. **An ablative analysis addressing this point can be found in Appendix D.1 of the original paper.** As indicated in Table 6, the verification module achieves a classification accuracy of approximately 70%, demonstrating the model's ability to identify erroneous answers. On the other hand, the correction module effectively reduces the errors identified during the verification stage by about 25%, confirming its effectiveness. These results are based on experiments conducted with the CosC-Code model at different sizes, specifically 7B and 13B, to ensure the reliability of our findings.
>
> ---
>
> ***W3: The main novelty of this paper is the verification/self-correction step. Without an extensive evaluation showing these capabilities have improved, it is hard to assess the effectiveness of the proposed method.***
>
> Thank you for your valuable comment. In fact, we have conducted several experiments to assess the effectiveness of the verification/self-correction step of our method, and we reported **the results in Tables 5 and Table 6** of the original paper. In Table 5, we observe that a single round of reasoning without self-correction leads to an approximately 7% decrease in accuracy. Furthermore, in Table 6, the verification module successfully identifies around 70% of erroneous answers, and the correction module reduces these errors by approximately 25%.
>
> We appreciate any specific suggestions you may have for further evaluating the self-correction capabilities and are open to exploring additional validation steps.

---

> ### Author Response · Authors · 2024-11-28
> **Rebuttal by Authors [3/3]**
>
> ***Q2: Compared to ToRA and looking at Table 5, it seems like the accuracy when using just one round of reasoning is lower than ToRA. Does that mean the data generated is worse than ToRA's when it comes to reasoning without any self-correction steps?***
>
> Thank you for your insightful observation regarding Table 5. The lower accuracy of our method compared to ToRA in a single round of reasoning does not necessarily indicate that the data generated by our method is inferior. Instead, it reflects the design focus of our approach, which emphasizes multi-round reasoning and self-correction.
>
> **Our method is explicitly optimized to leverage iterative reasoning, where the self-correction mechanism refines intermediate outputs over multiple rounds.** As a result, while the single-round performance may appear lower, the overall performance in multi-round scenarios demonstrates significant improvements. This trade-off highlights the unique strengths of our method in handling complex reasoning tasks, which rely on iterative refinement rather than single-pass outputs.
>
> We appreciate your point and believe it underscores the complementary nature of our approach to ToRA. As clearly shown in Table 4, the reasoning process of ToRA typically involves a single round of reasoning, with additional iterations occurring only in very rare cases.
>
> ---
>
> ***Q3: Because the models were trained on code, do they have an improved sense of code understanding?***
>
> Thank you for your thoughtful feedback. To address your question, we evaluate ToRA-Code and our CoSC-Code (both 7B models) on the MBPP dataset [2], which measures code understanding and generation. Both models are trained on mathematical datasets without code instructions.
>
> The results show that CoSC-Code achieves a pass@1 score of 37.6%, outperforming ToRA-Code, which achieves 30.8%. This indicates that **our CoSC approach enhances out-of-distribution generalization on code-related tasks compared to ToRA.** We attribute this improvement to the iterative reasoning and self-correction mechanisms embedded in our CoSC framework, which likely contribute to a stronger capacity for structured problem-solving and logical reasoning, even in domains like coding.
>
> [2] Jacob Austin, Augustus Odena, Maxwell Nye, Maarten Bosma, Henryk Michalewski, David Dohan, Ellen Jiang, Carrie Cai, Michael Terry, Quoc Le, Charles Sutton. Program Synthesis with Large Language Models. In 2021.

---

> > ### Author Response · Authors · 2024-12-02
> > **A Kind Reminder of Further Discussion**
> >
> > Dear Reviewer VQBN,
> >
> > Thank you once again for your valuable comments and suggestions, which have been extremely helpful to us. We have posted detailed responses to the concerns you raised and have included additional experimental results.
> >
> > We fully understand that this is a particularly busy period, and **we deeply appreciate it if you could take some time to provide further feedback on whether our responses address your concerns.** If there are any additional comments, we will try our best to address them promptly.
> >
> > Sincerely,\
> > Authors of Submission 7154

---

### Official Review · Reviewer_ediq · 2024-11-03

**Soundness:** 3
**Presentation:** 3
**Contribution:** 2
**Rating:** 6
**Confidence:** 3

**Summary:**

The paper presents a fine-tuning technique called Chain of Self-correction (CoSC) that embeds self-correction as an inherent ability of LLM. The method is specifically developed for LLM mathematics benchmarks - MATH and GSM8K. The paper proposes a specific format (program+output+verification+conclusion) for fine-tuning data. GPT4 is used first to get part of the fine-tuning data. This initial data is used to fine-tune a smaller model. After initial fine-tuning, the rest of the data is self-generated by a fine-tuned small model, and the model is further fine-tuned using the new data.

**Strengths:**

- The paper is well-written and easy to follow. I appreciate Figure 1, which summarizes prompts used in some recent related works.

- Compared to the prior work ToRA, the main difference is that the model is made to verify and confirm the answer. If the model deems that the output is incorrect then the model is made to repeat the whole procedure in multiple rounds. This improves the model’s accuracy on the benchmark by around 2%.

- Evaluation is quite comprehensive and considers many open-source and closed-source models

**Weaknesses:**

I can’t understand some of the main contributions of CoSC without some further clarification:

From what I can get from the evaluation in the paper - the second-best method in the evaluation is ToRA which converts a mathematical problem into Python code, evaluates it, and then uses CoT to infer the final solution. And the CoSC model improves on the ToRA model by around 2%.

How is ToRA prompted in this evaluation? If the ToRA models were not allowed to self-correct, is it possible to simply modify the prompt and use a similar prompt as CoSC (and possibly use some few-shot examples instead of fine-tuning) to allow it to self-correct? If not, then it would answer my main concern about whether collecting large amounts of data in two phases and fine-tuning in CoSC is actually necessary.

The algorithm 1 is referenced in line 310. I think this place is quite inappropriate for referring to the main algorithm. Consider moving the algorithm a little bit earlier in section 3.2.1.

I don’t understand what Table 4 for the ToRA code means without much information about the evaluation setup. From Figure 1, it seems that ToRA does not perform any self-correction, so what does the 0.1% cases for ToRA in Round=2 mean?

Line 142: “revolutionize the accuracy” - please remove revolutionize here

Line 215: what if the conclusion is never reached?

Grammar issues: Line 342, 344, 351, 363

Can you add more description to the appendix? Some parts of the appendix like Appendix B are quite unclear.

**Questions:**

As pointed out in weakness, my main question is about the prompt used for ToRA in Table 2.

Line 452: “Unlike these proprietary models, our CoSC performs the inference in a zero-shot manner without demonstration” -
I don’t understand this. How do the proprietary models considered in the evaluation use demonstrations?

---

> ### Author Response · Authors · 2024-11-25
> **Rebuttal by Authors [1/2]**
>
> Thank you for your valuable review and suggestions. Below we respond to the comments in **Weaknesses (W)** and **Questions (Q)**.
>
> ---
>
> ***W1 & Q1: How is ToRA prompted in this evaluation? Is it possible to simply modify the prompt to allow it to self-correct? Is fine-tuning in CoSC necessary?***
>
> Thank you for your insightful comment. To clarify, **ToRA is a fine-tuning-based method trained on data generated by GPT-4, as described in Line 86-87 and Line 437-438 of our original paper.** Unlike prompting methods, ToRA does not rely on instruction-based input during inference. After fine-tuning, it operates in a zero-shot manner without requiring external prompts or examples for reasoning tasks.
>
> In this evaluation, ToRA is directly applied to test tasks, without additional prompt modifications or few-shot examples. While it is theoretically possible to adapt ToRA by incorporating some prompts that simulates self-correction, this would fundamentally change its methodology and deviate from the original design.
>
> Our method, CoSC, is designed with self-correction as an intrinsic capability. Achieving this requires specific data collection and fine-tuning to enable robust iterative reasoning. This necessity arises from the fact that embedding self-correction as an inherent ability in LLMs **demands altering the model's underlying parameters, a process that extends beyond the scope of simple prompt adjustments.**
>
> ---
>
> ***W2: Consider moving the algorithm a little bit earlier in section 3.2.1.***
>
> Thank you for your helpful suggestion! Based on your feedback, we have moved the reference to Algorithm 1 to Line 260 in Section 3.2.1 in our revised paper to make it more accessible and to improve the flow of the presentation.
>
> ---
>
> ***W3: From Figure 1, it seems that ToRA does not perform any self-correction, so what does the 0.1% cases for ToRA in Round=2 mean?***
>
> Thank you for your valuable comment. According to the ToRA paper [1] and Figure 1 of our paper, ToRA generates a sequence consisting of a natural language guidance (r), a program (p), and an output (o) for a given question. This process is repeated until the model places its final answer within the “\boxed{}” symbol. The resulting trajectory is denoted as r₁p₁o₁...rₙ₋₁pₙ₋₁oₙ₋₁rₙ, where rₙ contains the answer.
>
> **For the case of Round=2 in ToRA, it implies that after the first round of reasoning (r₁p₁o₁), the answer generated by the model is insufficient.** As a result, the answer is not placed within the “\boxed{}” symbol. Consequently, ToRA continues the reasoning in the second round (r₂p₂o₂), and the final answer is placed within the “\boxed{}” symbol in r₃. For a detailed description of this generation process, please refer to Section 2.1 of the ToRA paper [1].
>
> One thing to note is that although the design of ToRA may suggest the use of multi-round reasoning, it does not explicitly possess the capability for multi-round inference, as clearly demonstrated in Table 4. Instead, the reasoning process of ToRA typically involves a single round of reasoning, with additional iterations occurring only in rare cases when the initial response is insufficient.
>
> [1] Zhibin Gou, Zhihong Shao, Yeyun Gong, Yujiu Yang, Minlie Huang, Nan Duan, Weizhu Chen, et al. Tora: A tool-integrated reasoning agent for mathematical problem solving. In ICLR, 2024.
>
> ---
>
> ***W4: Line 142: “revolutionize the accuracy” - please remove revolutionize here.***
>
> Thank you for your comment. We have revised the phrase in our updated version of the paper, removing the word "revolutionize" as suggested.
>
> ---
>
> ***W5: Line 215: what if the conclusion is never reached?***
>
> Thank you for your feedback. As stated in Line 367 of our paper, we set a maximum limit of three self-correction stages if a conclusion is not reached. We have further clarified this point in Line 241 of our revised paper.
>
> ---
>
> ***W6: Grammar issues.***
>
> Thank you for your careful reading. We have made the necessary revisions to address the grammar issues in the updated version of our paper.
>
> ---
>
> ***W7: Can you add more description to the appendix? Some parts of the appendix like Appendix B are quite unclear.***
>
> Thank you for your valuable feedback. In Appendix B, we provide an example of how our CoSC model generates corresponding answers in response to a query. Specifically, Line 1147-1148 contain the question posed to the CoSC model, while Line 1149-1210 present the answer generated by the model. We have clarified these details in the revised version of our paper for better understanding.

---

> ### Author Response · Authors · 2024-11-25
> **Rebuttal by Authors [2/2]**
>
> ***Q2: Unlike these proprietary models, our CoSC performs the inference in a zero-shot manner without demonstration - I don’t understand this. How do the proprietary models considered in the evaluation use demonstrations?***
>
> Thank you for your feedback. **In the evaluation of proprietary models such as GPT-4 and Gemini, the results we report are based on their respective official technical reports [2,3].** These reports indicate that the proprietary models often utilize few-shot prompting, where a small set of examples is provided within the prompt to guide their reasoning processes.
>
> In contrast, our method, CoSC, does not rely on such few-shot demonstrations during inference. Instead, CoSC leverages its intrinsic capabilities, enabled by the fine-tuning process, to perform zero-shot reasoning directly. This distinction highlights a fundamental difference in the inference paradigms: proprietary models benefit from carefully constructed few-shot examples to achieve optimal performance, whereas CoSC is designed to generalize robustly without such external demonstrations.
>
> [2] OpenAI. GPT-4 Technical Report. In 2023.\
> [3] Google Deepmind. Gemini 1.5: Unlocking multimodal understanding across millions of tokens of context. In 2024.

---

> > ### Comment · Reviewer_ediq · 2024-11-25
> > **Follow-up questions**
> >
> > > In the evaluation of proprietary models such as GPT-4 and Gemini, the results we report are based on their respective official technical reports [2,3]. These reports indicate that the proprietary models often utilize few-shot prompting
> >
> > In your Table can you add exact prompting method used for each evaluation? This is a key information for such an evaluation.
> >
> >
> > > In this evaluation, ToRA is directly applied to test tasks, without additional prompt modifications or few-shot examples. While it is theoretically possible to adapt ToRA by incorporating some prompts that simulates self-correction, this would fundamentally change its methodology and deviate from the original design.
> >
> > This makes me question the motivation of whole setup of this paper. Why is a restriction of inherently embedding self-correction justified if one could easily achieve the same level of accuracy with few in-context examples.

---

> > > ### Author Response · Authors · 2024-11-26
> > > **Thank you for your feedback**
> > >
> > > Thank you for your feedback! Below we respond to the remaining questions.
> > >
> > > ---
> > >
> > > ***Q1: Exact prompting method used for each evaluation.***
> > >
> > > We appreciate the reviewer’s valuable feedback. Table 2 in the original paper already includes an identifier "ZS", which denotes whether the LLMs are evaluated in a zero-shot inference setting without demonstrations. To clarify further, we summarize below the prompting methods employed for each evaluation:
> > >
> > > Proprietary Models:
> > > - GPT-4o [1]: Zero-shot CoT prompting for MATH; 8-shot CoT prompting for GSM8K.
> > > - GPT-4V [2]: 4-shot prompting for MATH; 5-shot CoT prompting for GSM8K.
> > > - GPT-4 and ChatGPT [3]: CoT prompting for MATH; 5-shot CoT prompting for GSM8K.
> > > - Gemini family [4]: 4-shot Minerva prompting for MATH; 11-shot prompting for GSM8K.
> > > - Claude family [5]: Zero-shot CoT prompting for both datasets.
> > > - PaLM-2 [6]: 4-shot CoT prompting for MATH; 8-shot CoT prompting for GSM8K.
> > >
> > > Open-Source Models:
> > > - LLaMA-2 [7] and Platypus-2 [8]: CoT prompting for both datasets.
> > > - CodeLLaMA [9]: Program-Aided Language (PAL) model prompting for both datasets.
> > > - LLaMA-2 SFT [10], LLaMA-2 RFT [10], WizardMath [11], MetaMath [12], ToRA [13], and our CoSC method: Fully zero-shot, requiring no demonstrations.
> > >
> > > We have included these information in Appendix C.2 of the revised paper for reference.
> > >
> > > [1] https://openai.com/index/hello-gpt-4o/. \
> > > [2] https://openai.com/index/gpt-4v-system-card/. \
> > > [3] GPT-4 Technical Report.\
> > > [4] Gemini 1.5: Unlocking multimodal understanding across millions of tokens of context.\
> > > [5] https://www.anthropic.com/news/claude-3-family. \
> > > [6] PaLM 2 Technical Report.\
> > > [7] Llama 2: Open Foundation and Fine-Tuned Chat Models.\
> > > [8] Platypus: Quick, cheap, and powerful refinement of llms.\
> > > [9] Code Llama: Open Foundation Models for Code.\
> > > [10] Scaling relationship on learning mathematical reasoning with large language models.\
> > > [11] Wizardmath: Empowering mathematical reasoning for large language models via reinforced evol-instruct.\
> > > [12] Metamath: Bootstrap your own mathematical questions for large language models.\
> > > [13] Tora: A tool-integrated reasoning agent for mathematical problem solving.
> > >
> > > ---
> > >
> > > ***Q2: Why is a restriction of inherently embedding self-correction justified if one could easily achieve the same level of accuracy with few in-context examples.***
> > >
> > > Thanks for your insightful comments. As suggested by the reviewer, we have conducted an additional experiment using the CoSC prompts for evaluation on ToRA-Code-7B. The results on the MATH and GSM8K datasets are shown in the table below.
> > >
> > > |  | MATH | GSM8K |
> > > |-----------|------|------|
> > > | ToRA-Code-7B  | 44.6 | 72.6 |
> > > | ToRA-Code-7B with CoSC prompt | 42.8 | 68.0 |
> > > | CoSC-Code-7B (ours) | 47.6 | 74.7 |
> > >
> > > As shown in the table, applying CoSC prompting to ToRA not only fails to outperform the original ToRA model but also results in a decline in performance. As clearly demonstrated in Table 4 of the original paper, ToRA inherently lacks the robust multi-round reasoning capabilities needed for effective self-correction. When CoSC prompting is applied, it introduces complexity that the model is ill-equipped to handle, leading to confusion and errors in the iterative process. Similarly, during the development of the CoSC algorithm, we also attempt to apply self-correction prompts to the base CodeLLaMA model. However, this approach did not yield good performance and was significantly lower than the previous state-of-the-art results in open-source models. **This led us to adopt a fine-tuning strategy instead.**
> > >
> > > In contrast, our CoSC model, which integrates self-correction as an inherent capability via fine-tuning, achieves superior results on both datasets. These findings suggest that for open-source LLMs, few-shot prompting alone is insufficient to effectively enable self-correction. The lack of significant gains from prompting further underscores the limitations of relying solely on in-context examples. **Therefore, we argue that embedding self-correction as an inherent capability through fine-tuning is essential for truly endowing LLMs with robust self-correction abilities.**
> > >
> > > Moreover, by integrating self-correction directly into the training process, our approach allows models to perform self-correction autonomously in a zero-shot setting during inference, eliminating the need for external feedback or few-shot demonstrations. **This self-correction mechanism enables even weaker LLMs to achieve significant improvements in mathematical reasoning—enhancements that are unattainable through prompting methods alone.** Additionally, our CoSC framework is **open-source,** making these advancements accessible to the broader research community. We believe this represents a pivotal step toward democratizing advanced reasoning capabilities and fostering further innovation.

---

> > > > ### Comment · Reviewer_ediq · 2024-11-26
> > > >
> > > > Thank you for the response.
> > > >
> > > > I would recommend the authors to include the key details on the prompting method in Table 2 if possible rather than in the Appendix.
> > > >
> > > > >  we have conducted an additional experiment using the CoSC prompts for evaluation on ToRA-Code-7B. The results on the MATH and GSM8K datasets are shown in the table below.
> > > >
> > > > Can you include the details of the additional experiment and the exact prompt used in the Appendix?

---

> > > > > ### Author Response · Authors · 2024-11-27
> > > > > **Thank you for your feedback**
> > > > >
> > > > > Thank you for your feedback! Below we respond to the remaining questions.
> > > > >
> > > > > ---
> > > > >
> > > > > ***Q1: I would recommend the authors to include the key details on the prompting method in Table 2 if possible rather than in the Appendix.***
> > > > >
> > > > > Thank you for your valuable suggestion. We have included the key details of the prompting methods for each evaluation **in Table 2 of the revised paper.**
> > > > >
> > > > > ---
> > > > >
> > > > > ***Q2: Can you include the details of the additional experiment and the exact prompt used in the Appendix?***
> > > > >
> > > > > Thank you for your valuable suggestion. We have included the details of the additional experiment and the corresponding prompts **in Appendix D.4 of the revised paper.** The prompts used are the same as those used for CoSC seeding data generation in Appendix A.

---

> ### Comment · Reviewer_ediq · 2024-11-30
>
> Thank you for your response. I am updating my overall score to 6.
>
> I would encourage the authors to explore incorporating evaluation methods that incentivize "self-correction" through the use of a large number of in-context examples, rather than relying solely on fine-tuning. My understanding is that fine-tuning a model for self-correction might reduce its performance on other tasks. If this understanding is incorrect, I would appreciate clarification.
> Additionally, I would like to see a more detailed discussion on the future directions of this research. For instance, How does this research fit in the context of more recent models trained for reasoning such as GPT4-o1, DeepSeek R1, or Qwen QwQ?

---

> > ### Author Response · Authors · 2024-12-01
> > **Thank you for your support and raising the score**
> >
> > Thank you very much for your thoughtful feedback and for updating your overall score. We greatly appreciate your valuable suggestions and would like to address your points as follows.
> >
> > ---
> >
> > ***Q1: I would encourage the authors to explore incorporating evaluation methods that incentivize "self-correction" through the use of a large number of in-context examples, rather than relying solely on fine-tuning. My understanding is that fine-tuning a model for self-correction might reduce its performance on unrelated tasks. If this understanding is incorrect, I would appreciate clarification.***
> >
> > Thank you for your insightful suggestion. We agree that prompting, using a large number of in-context examples, can be a promising approach to complement our fine-tuning-based CoSC method. Besides, we also agree that fine-tuning (SFT) a model for a specific task can lead to catastrophic forgetting, where the model’s performance on unrelated tasks may deteriorate. However, it is important to emphasize that the goal of our CoSC approach is to embed self-correction as an inherent capability in LLMs, **specifically to enhance mathematical reasoning.** For a specialized LLM specifically for mathematical reasoning, we think that the performance improvement on mathematical tasks outweighs the potential drawbacks on unrelated tasks. Furthermore, we posit that, with careful datasets design, such as involving datasets from other tasks, the impact of catastrophic forgetting can be mitigated, allowing the model to maintain performance across a broader range of tasks. However, we recognize the value of incorporating in-context examples to incentivize self-correction, in addition to fine-tuning. We view this as an important direction for future research and look forward to exploring it in our ongoing work.
> >
> > ---
> >
> > ***Q2: Additionally, I would like to see a more detailed discussion on the future directions of this research. For instance, How does this research fit in the context of more recent models trained for reasoning such as GPT4-o1, DeepSeek R1, or Qwen QwQ?***
> >
> > Thank you for your insightful suggestion! We appreciate your interest in the future directions of our research. We plan to explore several key areas to further advance our work:
> >
> > - **Extending to Broader Domains.** Our CoSC approach introduces a structured, step-by-step framework that enables LLMs to identify errors, generate corrections, and iteratively refine their outputs with greater accuracy. While we have demonstrated its effectiveness in mathematical reasoning tasks, we believe that the self-correction mechanisms can be extended to a broader range of domains. In future work, we aim to adapt CoSC for tasks such as general reasoning, code generation, and multimodal applications. This expansion will allow us to explore new areas where self-correction can further enhance LLM performance.
> > - **Advancing Prompting Techniques.** In addition to fine-tuning, we will investigate methods to prompt LLMs for self-correction, leveraging their inherent abilities. This includes manually designing effective prompts, automatically selecting high-quality few-shot examples for various tasks, and crafting concise prompts to facilitate self-correction. We believe that refining in-context examples to prompt LLMs for self-correction will be a valuable avenue for future research. Furthermore, with the emergence of state-of-the-art models such as GPT-4-o1, DeepSeek R1, and Qwen QwQ, there is an increasing need to develop adaptive, context-aware prompting techniques, which can better align with the advanced capabilities of these models and enhance their self-correction potential.
> > - **Open-Sourcing Code, Data, and Models.** In our commitment to advancing research in the field, we will open-source the code, data, and models related to this work. Once the paper is accepted, all associated resources will be made publicly available. We believe that this openness will foster engagement from the broader research community and support the development of more flexible and versatile reasoning systems, allowing CoSC to be applied across a wider range of applications.
> >
> > We will incorporate these discussions into the next version of our paper and continue to explore these exciting future directions in our ongoing research.

---

### Author Response · Authors · 2024-11-25
**General Response**

We would like to thank the AC for organizing the review of our paper. We found all the reviews and comments extremely helpful. We have addressed all the concerns raised by the reviewers and have revised the paper accordingly to properly address any questions raised.

We would also like to thank all the reviewers for taking the time to review our paper and provide valuable feedback:
- We would like to thank Reviewer ediq for recognizing the clarity of our paper, the usefulness of Figure 1 in summarizing related prompts, and for highlighting the improvement in accuracy enabled by our CoSC mechanism.
- We are grateful to Reviewer VQBN for appreciating our framework for enhancing self-correction abilities and for acknowledging the potential future value of this resource for training models.
- We sincerely thank Reviewer KVMJ for commending our two-phase fine-tuning approach, which leverages self-correction to reduce reliance on expensive API calls, and for highlighting the strength of our methodology in enabling models to act as their own teacher.
- We also thank Reviewer gRDy for recognizing the cost-efficiency and effectiveness of our two-phase fine-tuning approach, the breadth of our evaluation across various model sizes, and the clarity and structure of our paper.



We address each reviewer’s comments individually below. We have worked hard to address your concerns and hope you find our responses informative. If you feel our comments have not sufficiently addressed your concerns, we would love to discuss them with you further. We have also uploaded a **Paper Revision** for your consideration.

---

### Author Response · Authors · 2024-11-30
**A Kind Reminder of Further Discussion**

Dear Reviewers,

We would like to express our sincere gratitude for taking the time to review our paper. We greatly appreciate your insightful comments and are truly thankful for your efforts. We have worked diligently to address all of the concerns you raised, and we hope that our responses have sufficiently resolved any remaining issues. **If you have any further questions or require clarification, we would be more than happy to discuss them with you.** Your feedback is invaluable to us.

Sincerely,\
Authors of Submission 7154

---

### Meta-Review · Area_Chair_BfjP · 2024-12-27

**Metareview:**

This paper introduces Chain of Self-Correction (CoSC), a fine-tuning mechanism enabling LLMs to iteratively self-correct by generating programs, executing them, verifying outputs, and refining or finalizing answers. It uses a two-stage data synthesis approach: GPT-4 generates initial data to train a seed LLM, which then generates additional data to fine-tune itself. Experiments show that CoSC outperforms existing prompting and fine-tuning methods on MATH and GSM8K across proprietary and open-source LLMs.

Most reviewers and the AC generally acknowledge that the proposed method is well-motivated, with comprehensive experiments to support its claims. However, concerns are raised regarding the novelty of the approach. The method integrates several well-established ideas, such as tool use, verification, and iterative reasoning, without introducing significant conceptual innovation. Similarly, the data synthesis strategy combines well-known techniques like data distillation and expert iteration. While the improved performance on math datasets is notable, the gains over prior work, such as ToRA, appear marginal.

Overall, while the paper presents a promising approach and demonstrates incremental improvements, the lack of substantial novelty and limited performance gains lead me to consider this a borderline submission.

**Additional Comments On Reviewer Discussion:**

During the rebuttal period, reviewers raised several concerns regarding certain claims and the experimental setup presented in the paper, most of which were adequately addressed by the authors. However, concerns regarding the limited novelty and marginal performance improvements remain.

---

### Decision · Program_Chairs · 2025-01-22

Reject